# Cell volume controlled by LRRC8A-formed volume-regulated anion channels fine-tunes T cell activation and function

Yuman Wang[1], Zaiqiao Sun[2], Jieming Ping[1], Jianlong Tang[1], Boxiao He[2], Teding Chang[3], Qian Zhou[4], Shijie Yuan[1], Zhaohui Tang[3], Xin Li[5], Yan Lu [6], Ran He[1], Ximiao He[4], Zheng Liu [7]✉, Lei Yin [2]✉ & Ning Wu [1,7,8,9]✉

Biosynthesis drives the cell volume increase during T cell activation. However, the contribution of cell volume regulation in TCR signaling during T lymphoblast formation and its underlying mechanisms remain unclear. Here we show that cell volume regulation is required for optimal T cell activation. Inhibition of VRACs (volume-regulated anion channels) and deletion of leucine-rich repeat-containing protein 8A (LRRC8A) channel components impair T cell activation and function, particularly under weak TCR stimulation. Additionally, LRRC8A has distinct influences on mRNA transcriptional profiles, indicating the prominent effects of cell volume regulation for T cell functions. Moreover, cell volume regulation via LRRC8A controls T cell-mediated antiviral immunity and shapes the TCR repertoire in the thymus. Mechanistically, LRRC8A governs stringent cell volume increase via regulated volume decrease (RVD) during T cell blast formation to keep the TCR signaling molecules at an adequate density. Together, our results show a further layer of T cell activation regulation that LRRC8A functions as a cell volume controlling "valve" to facilitate T cell activation.

Though mammalian cell volume/size differs enormously among different cell types, it is tightly constrained to a limited range within a specific cell type. Cell volume adaptation in response to variations of osmolarity is a fundamental and primitive cellular function in various physiological processes[1,2]. Uncontrolled cell volume impairs cell signaling transduction and cell cycle arrest, leading to senescence and other cellular abnormalities[3,4]. Proliferative and metabolic activities contribute substantially to normal cell volume regulation[1,5,6]. Immune cells can preserve their size to an approximately constant range in response to the alteration of osmolarity in the medium[7–10]. However, the physiological relevance of immune cell volume regulation remains puzzled in the body with nearly constant osmotic tonicity.

T cell development, activation, and function in response to antigens require proper TCR signals following the recognition of the peptide-MHC complex (pMHC)[11,12]. Upon initiation of TCR signaling, naïve T cells rapidly exit from a basal metabolic, transcriptional and

[1]Department of Immunology, School of Basic Medicine, Tongji Medical College, Huazhong University of Science and Technology, Wuhan, China. [2]State Key Laboratory of Virology, Hubei Key Laboratory of Cell Homeostasis, College of Life Sciences, Renmin Hospital of Wuhan University, Wuhan University, Wuhan, China. [3]Department of Traumatic Surgery, Tongji Trauma Center, Tongji Hospital, Tongji Medical College, Huazhong University of Science and Technology, Wuhan, China. [4]Department of Physiology, School of Basic Medicine, Tongji Medical College, Huazhong University of Science and Technology, Wuhan, China. [5]Medical Research Center, Guangdong Provincial People's Hospital, Guangdong Academy of Medical Sciences, Guangzhou, China. [6]Department of Clinical Immunology, The Third Affiliated Hospital of Sun Yat-sen University, Guangzhou, China. [7]Department of Otolaryngology-Head and Neck Surgery, Tongji Hospital, Tongji Medical College, Huazhong University of Science and Technology, Wuhan, China. [8]Cell Architecture Research Center, Tongji Medical College, Huazhong University of Science and Technology, Wuhan, China. [9]The First Affiliated Hospital of Anhui Medical University, Institute of Clinical Immunology, Anhui Medical University, Hefei, China. ✉e-mail: zhengliuent@hotmail.com; yinlei@whu.edu.cn; wuning118@gmail.com

translational quiescent status and switch to robust anabolic metabolism[13,14]. During activation, intracellular osmolytes dramatically augment due to the ions' influxes, biosynthesis including RNA transcription, protein synthesis, and other small organic metabolites production. The accumulation of these intracellular osmolytes creates higher osmolarity in the cell, which results in water influxes and the enlargement of T cell volume. This osmolarity-driven cell swelling phenomenon is well-known as T cell blast. This process is reminiscent of leukocyte swelling suspended in the hypotonic buffer, but temporally stretched[8]. However, its underlying mechanism and physiological function remain unknown.

Cellular osmotic swelling is regulated by the volume-regulated anion channels (VRACs)[15]. In the presence of VRACs, swollen cells can actively control the cell volume by regulated volume decrease (RVD) and modulate innate immune response[16]. Recently, the leucine-rich repeat-containing protein 8 (LRRC8) family has been identified as the critical VRAC component in mammals during hypo-osmotic swelling[17,18]. LRRC8 family consists of five members, from LRRC8A to E. As the only obligatory subunit of VRAC, LRRC8A forms a hexamer with other LRRC8 family members and is activated by sensing low ionic strength to facilitate the RVD process[19,20]. Besides the Cl⁻ channel function, LRRC8A-formed VRAC has been reported to be the transporter of various small molecules, such as 2'3' cGMP-AMP (cGAMP), cisplatin, GABA and taurine, and participate in inflammasome activation and innate immunity[21–28].

*LRRC8* gene mutation was initially described in human agammaglobulinemia[29]. In a germline knockout (KO) mouse model, LRRC8A is found to be essential in T cell development[30]. However, the VRAC function of LRRC8A seems to be dispensable for T cell development, as shown by the spontaneous mouse mutant (*ebo/ebo*) with Cl⁻ channel activity defects[31]. Recently, the non-essential component of VRAC LRRC8C has been reported to suppress CD4⁺ T cell function via STING-p53 signaling[32].

In this study, we generated a T-cell-specific *Lrrc8a* KO mouse model (*Lrrc8a^{f/f, Cd4-Cre}*) to determine the role of cell volume regulation mediated by LRRC8A intrinsically in T cell activation and function. During T cell blasts, the VRAC inhibition or genetic depletion of LRRC8A impaired T cell activation, cytokine production, and proliferation, particularly at weak TCR signal strength. In addition, RNA sequencing of OT-I CD8⁺ T cells activated by different TCR agonists revealed distinctive transcriptional profiles at various TCR signal strengths in WT and LRRC8A KO cells, implying T cell activation and function can be substantially modulated by LRRC8A. Furthermore, in an acute lymphocytic choriomeningitis virus (LCMV) Armstrong infection, CD8⁺ T cell-mediated antiviral immunity is abnormal in the absence of LRRC8A. Moreover, by TCR sequencing (TCR-seq) of thymocytes, we found more diversified TCR repertoire in *Lrrc8a^{f/f, Cd4-Cre}* mice, presumably due to a weaker TCR signaling by self-peptides during selection in the thymus. Finally, using a hypotonic environment to mimic the osmotic swelling elicited by osmolyte accumulation during T cell activation, we provided evidence that the proximal signal defects in LRRC8A-deficient cells resulted from decreased molecular density and interactions due to the cell volume enlargement. During the T cell blast, LRRC8A-dependent RVD endows T cells with molecular density assurance to counteract the decay of the TCR signal. Together, our work demonstrated that T cell activation and function would not be optimal without appropriate cell volume regulation, elucidated the mechanism of cell volume regulation by LRRC8A during T cell activation, and discovered its physiological role in shaping T cell repertoire and antiviral immunity.

## Results

### Impaired T cell activation and function by VRAC inhibition

To address the role of cell volume regulation in T cell activation during T cell blast, we first verified the T cell volume increase upon stimulation both in vivo and in vitro. By administering ovalbumin intraperitoneally (ip), OT-I CD8⁺ T cells and OT-II CD4⁺ T cells adoptively transferred into C57BL/6J (referred to B6 hereafter) mice were activated, shown by the induction of CD69 expression (Supplementary Fig. 1a). OT-I CD8⁺ T cells and OT-II CD4⁺ T cells express TCR recognizing ovalbumin peptides (OVA₂₅₇₋₂₆₄ and OVA₃₂₃₋₃₃₉) bound to MHC H2-K^b and I-A^b, respectively. Under isotonic conditions, cell volume is proportional to FSC on flow cytometry[33]. Compared with endogenous unstimulated T cells in B6 recipients, both activated OT-I CD8⁺ T cells, and OT-II CD4⁺ T cells have significant augmentation of cell volume, detected by FSC (Fig. 1a). Similarly, using LCMV strain Armstrong to infect the B6 recipient mice transferred with P14 splenocytes whose CD8⁺ T cells bear TCR recognizing LCMV-specific epitope gp33 presented by H2-D^b, activated P14 CD8⁺ T cells are bigger than CD8⁺ T cells from B6 recipients at day three pi (Fig. 1b). Besides the antigen-specific activation of T cells in vivo, antibody crosslinking (anti-CD3ε) or mitogen concanavalin A (ConA) in vitro stimulated T cells both exhibited cell volume increase (Supplementary Fig. 1b, c). The increase of T cell size upon activation was further confirmed by a confocal microscope (Supplementary Fig. 1d).

Cell swelling is tightly regulated by VRACs[1,15,17,18]. To determine the contribution of controlled cell volume increase during T cell blast in activation, we tested whether inhibition of VRACs-mediated cell swelling through the well-established VRAC inhibitor, 4-(2-Butyl-6,7-dichloro-2-cyclopentyl-indan-1-on-5-yl) oxobutyric acid (DCPIB)[17,18,28], which abrogates the RVD process during cell swelling, affects T cell activation. Incubation of OT-I CD8⁺ T cells with DCPIB has no cellular toxicity under our experimental settings (Supplementary Fig. 1e). Using OVA peptide SIINFEKL (N4, OVA₂₅₇₋₂₆₄) to activate OT-I CD8⁺ T cell in vitro, the upregulation of early T cell activation markers CD69 and CD25 both were significantly attenuated when VRACs were blocked by DCPIB, particularly more evident at the lower doses of N4 peptide (e.g., 1 and 10 pM) (Fig. 1c). Despite unchanged percentages of CD69 and CD25 expression at the higher dose of N4 peptide (e.g., 10² and 10³ pM), their molecular densities (MFI) on the cell membrane were reduced upon inhibition of VRACs (Fig. 1d). Consequently, less activated OT-I CD8⁺ T cells, due to the VRAC blockade by DCPIB, behaved less blasted (Fig. 1e). Similarly, DCPIB inhibited splenic CD4⁺ and CD8⁺ T cells activation and blast with antigen-independent stimulation of T cells by anti-CD3ε crosslinking and ConA (Fig. 1f and Supplementary Fig. 1f). These data demonstrated that T cell activation and blast are impeded when VRAC is malfunctional, implying T cell volume control is not only an adjoint phenomenon during activation but also actively regulates T cell activation.

T cell activation relies on TCR signaling. We found the phosphorylation of S6 (p-S6^{S240/244}) and MAPK (p-ERK1/2^{T202/Y204}) upon TCR stimulation was restrained in T cells from either OT-I or B6 mice when VRAC activity was abrogated by DCPIB (Fig. 1g and Supplementary Fig. 1g, h). Of note, S6 phosphorylation is known to be implicated in protein synthesis and cell size regulation[34]. In addition, T cell activation immediate-early response protein Nur77 (NR4A1) was less expressed in the absence of VRAC as well (Fig. 1h). In line with this, the blockade of VRACs using DCPIB suppressed T cell proliferation upon activation, and the difference was more pronounced with low doses of N4 peptide (Fig. 1i and Supplementary Fig. 1i). Concomitantly, T cell effector function, such as cytokine production of TNFα and IFNγ, was severely compromised upon activation when VRACs were arrested (Fig. 1j and Supplementary Fig. 1j). Moreover, naïve T cells exit from quiescence to proliferation and shift to dramatic biosynthesis upon activation[13,14]. During this transition, the nutrient transporters, including CD71 and CD98, are usually upregulated[13,14]. We also found the induction of CD71 and CD98 after OT-I CD8⁺ T cell activation, which was lower in VRAC-inhibited T cells (Fig. 1k). Together, normal VRAC function is required by optimal T cell activation, proliferation, and function during blast in CD4⁺ and CD8⁺ T cells, regardless of antigen-specificity.

## LRRC8A is essential for optimal T cell activation

LRRC8 family was previously identified as VRACs to control the RVD process in hypotonic cell swelling[17,18]. To exclude the possible off-target effects by DCPIB[35–38], we sought to interrupt the VRAC function genetically. As the essential component of LRRC8-formed VRACs, LRRC8A was constitutively expressed in both thymocytes and splenic T cells and remained at the same levels after activation and expansion by IL-2 in both splenic CD4[+] and CD8[+] T cells in vitro (Supplementary Fig. 2a–c), suggesting that LRRC8A is ready to function during T cell blast in response to activation. To explore the role of LRRC8A in T cell activation, we generated *Lrrc8a* floxed (*Lrrc8a*[f/f]) mice by CRISPR-Cas9 technology (Supplementary Fig. 2d). T-cell-specific knockout (KO) mice were obtained by crossing *Lrrc8a*[f/f] with *Cd4-Cre* mice (*Lrrc8a*[f/f, Cd4-Cre]), and the depletion of LRRC8A was verified in thymocytes and purified

splenic CD4[+] and CD8[+] T cells (Fig. 2a). LRRC8A T-cell-specific KO mice breed normally, and their T cells develop normally in both the thymus and spleen (Supplementary Fig. 2e–h). Consistent with previous reports that LRRC8A is an obligatory component of VRAC, LRRC8A-deficient thymocytes lost their volume regulation ability under hypotonic conditions, exhibiting bigger cell volume than their WT counterparts owing to loss of RVD (Supplementary Fig. 2i).

To determine the role of LRRC8A-mediated cell volume regulation during CD8[+] T cell blast in antigen-specific T cell response, we crossed the *Lrrc8a* T cell-specific KO mice to OT-I transgenic background (*Lrrc8a*[f/f; OT-I] as the control, and *Lrrc8a*[f/f, Cd4-Cre; OT-I] as the KO). Using different doses of SIINFEKL peptide (N4 peptide, OVA[257-264]), ranging from 0.1 pM to 10[3] pM, to activate splenic OT-I CD8[+] T cells, loss of LRRC8A resulted in a significant deficit of activation, especially

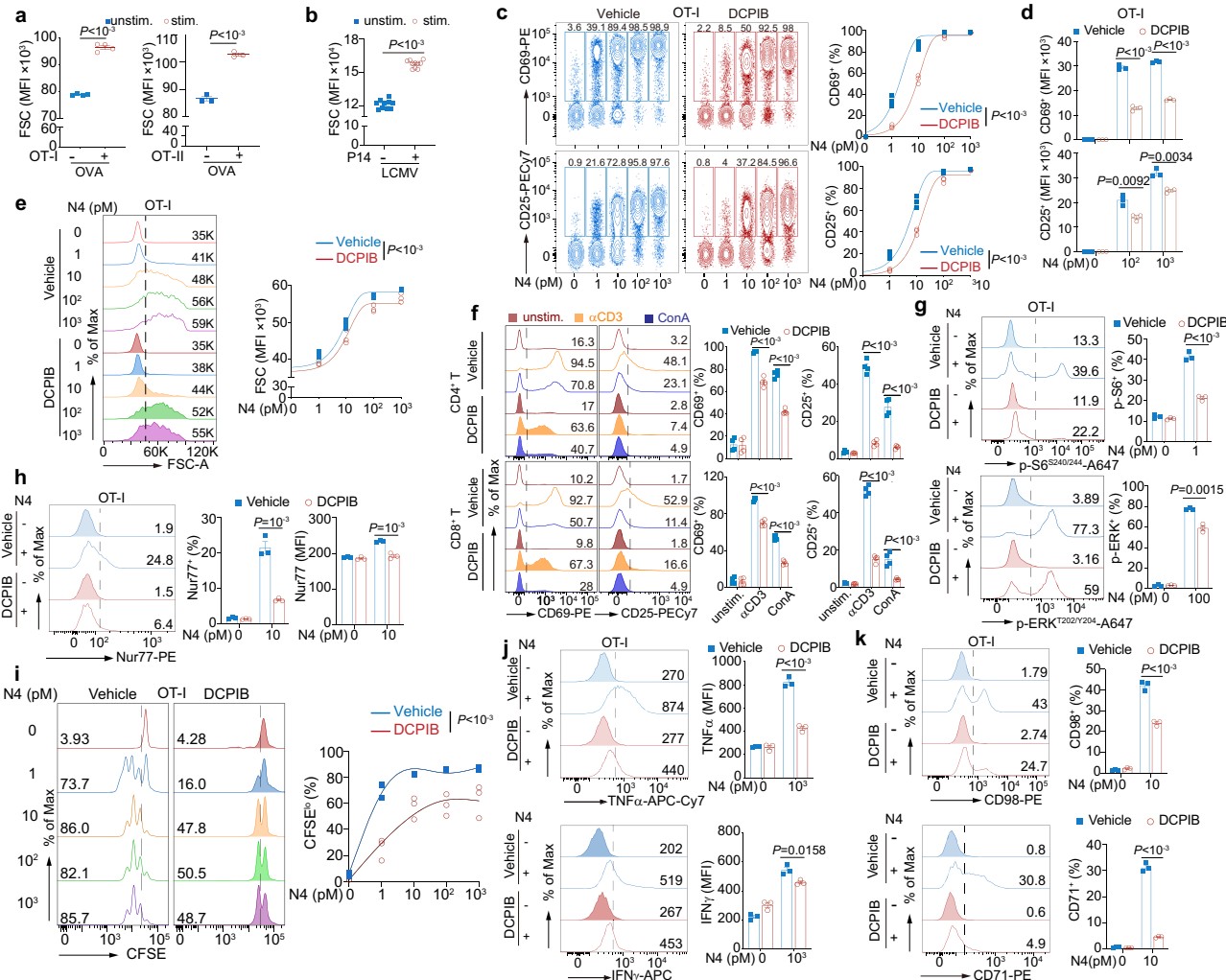

**Fig. 1 | T cell activation and function require VRACs. a, b** T cell size upon activation in vivo evaluated by forward scatter (FSC) on flow cytometry. OT-I and OT-II (labeled by CellTrace Violet before transfer) or P14 cells (CD45.1[+]) obtained from spleen after activation by antigen (OVA, ovalbumin or LCMV Armstrong) at indicated time (day 2 after OVA injection for (**a**) day 3 after infection for (**b**)) (OT-I, *n* = 3 mice/group; OT-II, *n* = 4 mice/group; P14, *n* = 10 mice/group). **c, d** OT-I CD8[+] T cell activation monitored by CD69 and CD25 expression. Splenic OT-I CD8[+] T cells were stimulated by a variety of doses of N4 peptide ex vivo for 6 h, ±DCPIB (25 μM) gated on TCRβ[+] CD8[+] Vα2[+] (*n* = 3 replicates/mouse). **e** Cell size of T cells activated as in panel **c**, measured by flow cytometry 24 h after activation (*n* = 3 replicates/mouse). **f** T cell activation assessed by CD69 and CD25 expression. Splenic CD4[+] and CD8[+] T cells were activated by anti-CD3ε (0.1 μg/ml) or ConA (1 μg/ml) for 6 h ex vivo, with or without DCPIB (25 μM) (*n* = 4 mice/group). **g–k** Signaling, activation,

proliferation, cytokine production, and nutrition markers in OT-I CD8[+] T cell activated by N4 peptide at indicated concentration in vitro, ±DCPIB, examined by intracellular flow cytometry. Phosphorylation of S6 (p-S6[S240/244]) (stimulated for 6 h) and MAPK (p-ERK[T202/Y204]) (stimulated for 2 h) (**g**), Nur77 (stimulated for 6 h) (**h**), proliferation shown by CFSE dilution (activated for 48 h) (**i**), TNFα and IFNγ production (activated for 6 h) (**j**), CD98 and CD71 expression (activated for 6 h) (**k**) (*n* = 3 replicates/mouse for **g–k**). Representative flow cytometry plots were shown on the left and the corresponding quantification on the right (**c, e–k**). Data are representative of two (**a, h–k**), three (**c–e, g**), or more (**b, f**) independent experiments. Unpaired two-sided *t*-test was used in (**a, b, d, f–h, j, k**). Two-way ANOVA with Bonferroni's post-hoc test were in (**c, e, i**). Data are presented as mean ± SEM. Source data are provided as a Source Data file.

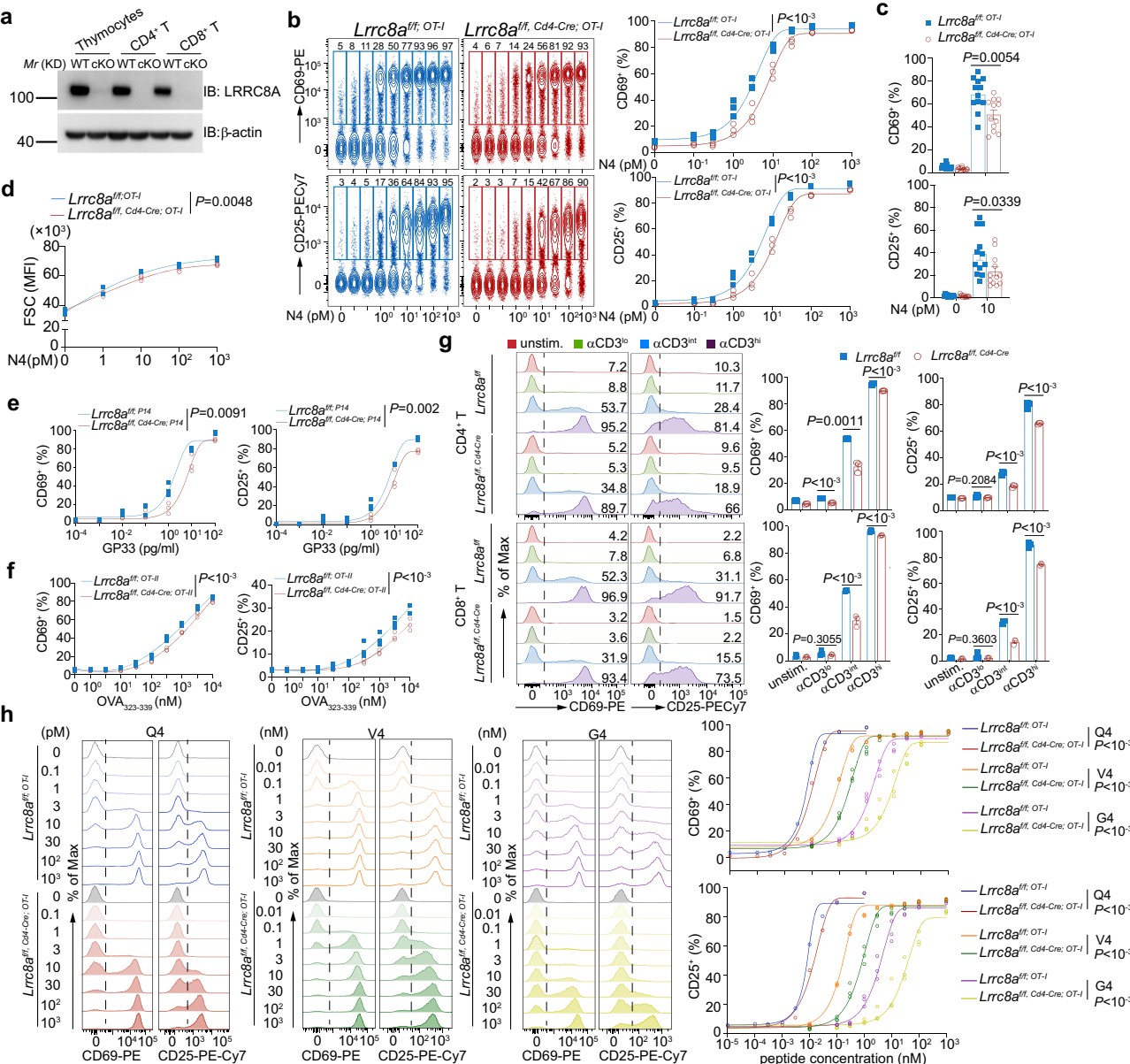

**Fig. 2 | LRRC8A is essential for optimal T cell activation. a** Immunoblot of LRRC8A in thymocytes, purified splenic CD4+, or CD8+ T cells from WT (*Lrrc8a^f/f*) and conditional KO (cKO, *Lrrc8a^f/f,Cd4-Cre*) mice. **b–c** Surface staining of CD69 and CD25 quantified by flow cytometry. OT-I CD8+ T cells from *Lrrc8a^f/f, OT-I* or *Lrrc8a^f/f, Cd4-Cre; OT-I* splenocytes were administered by N4 peptide ex vivo for 6 h at indicated concentrations, gated on TCRβ+ CD8+ Vα2+ (*n* = 3 replicates per mouse). Representative experiment in (**b**), and summarized biological replicates in (**c**) (*n* = 12 mice/group). **d** T cell blast was monitored by FSC on flow cytometry 24 h post-stimulation ex vivo. Splenic OT-I CD8+ T cells from *Lrrc8a^f/f; OT-I* or *Lrrc8a^f/f; Cd4-Cre; OT-I* mice were activated by N4 peptide at indicated concentrations (*n* = 3 replicates per mouse). **e, f** Expression of CD69 and CD25 on OT-II and P14 cells. WT and LRRC8A KO T cells from OT-II (**f**) or P14 background (**e**) were activated by a variety of doses of OVA_{323-339} (**f**) or GP33 (**e**) peptide ex vivo for 6 h. OT-II CD4+ T cells and P14 CD8+ T cells were gated on TCRβ+

CD4+ Vα2+ and CD45.1+ CD8+ Vα2+, respectively (*n* = 3 replicates/mouse for (**e**) and (**f**)). **g,** CD69 and CD25 expression on splenic CD4+ or CD8+ T cells from WT (*Lrrc8a^f/f*) and conditional KO (cKO, *Lrrc8a^f/f,Cd4-Cre*) mice were activated by anti-CD3ε ex vivo for 6 h. Antibody concentrations: αCD3lo (0.001 μg/ml), αCD3int (0.01 μg/ml), αCD3hi (0.1 μg/ml). Gated on CD4+ or CD8+ T cells (*n* = 3 replicates per mouse). **h** OT-I CD8+ T cell activation by N4 peptide variants Q4, V4 and G4, shown by CD69 and CD25 (*n* = 3 replicates per mouse). Representative flow cytometry plots of CD69 and CD25 were shown on the left and their quantification was on the right for **b, g, h.** Data are representative of two (**a, b, g**), three (**h**), or more (**d–f**) independent experiments. Two-way ANOVA with Bonferroni's post-hoc test was in (**b, d, e, f, h**). Unpaired two-sided *t* test was in (**c, g**). Data are presented as mean ± SEM. Source data are provided as a Source Data file.

at low and intermediate doses of N4 peptide, monitored by CD69 and CD25 expression (Fig. 2b, c). Accordingly, T cells blasted after activation, and LRRC8A-deficient T cells were more petite than WT T cells (Fig. 2d). Similarly, we confirmed this result in antigen-specific T cells in response to GP33 (a peptide from the lymphocytic choriomeningitis virus, glycoprotein gp33–41, presented by the MHC-I molecule H-2D^b P14 CD8+ T cell) and OVA_{323-339} peptide (a peptide from ovalbumin, presented by MHC-II molecule I-A^b to OT-II CD4+ T cell) respectively

(Fig. 2e, f). Moreover, in an antigen-independent T cell activation, crosslinking with anti-CD3ε antibody also showed reduced activation in LRRC8A KO splenic CD4+ and CD8+ T cells at various concentrations but more pronounced at unsaturated antibody dosage (about half of the control cells at intermediate dose) (Fig. 2g and Supplementary Fig. 3a–d). Consequently, T cells were less blasted in both CD4+ and CD8+ T cells (Supplementary Fig. 3e,f). These aforementioned results revealed the involvement of LRRC8A in T cell activation, and the

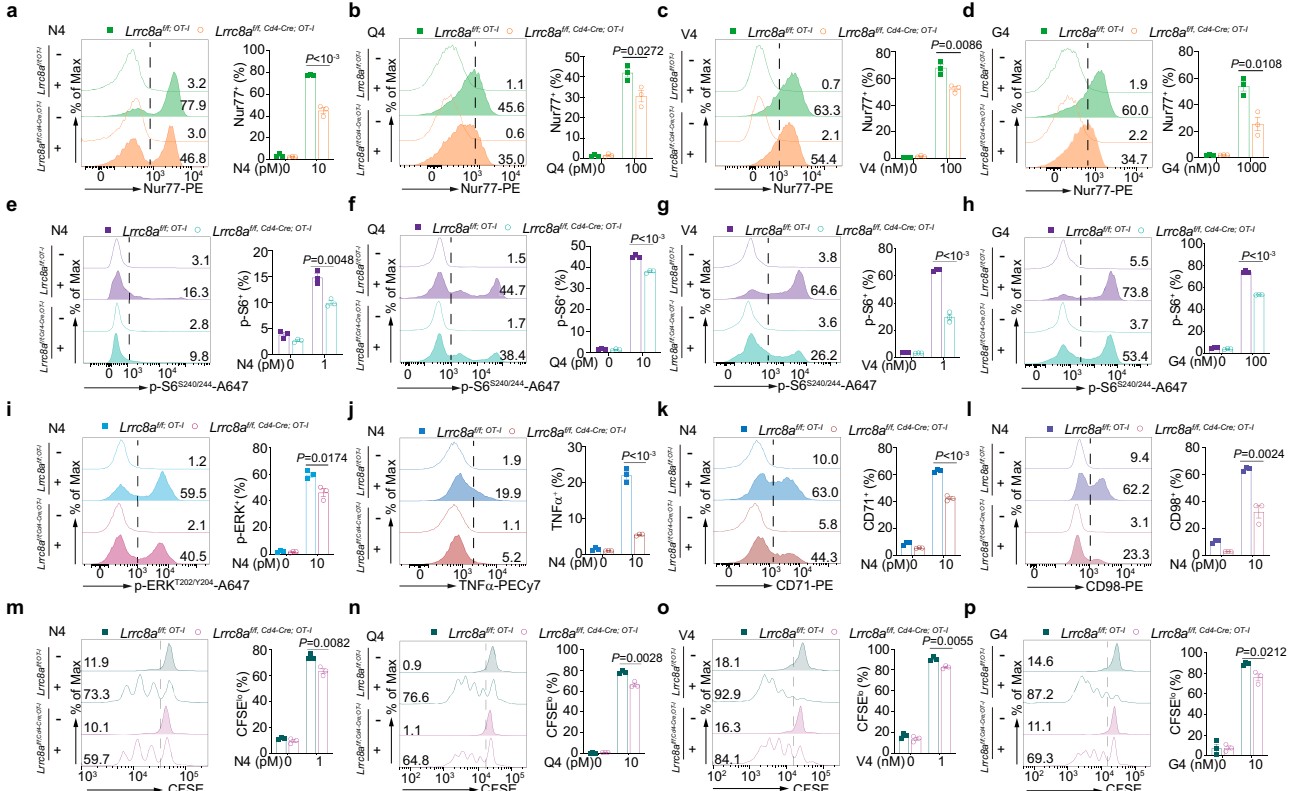

**Fig. 3 | Defective T cell signaling and function in the absence of LRRC8A.** OT-I CD8⁺ T cells from WT (*Lrrc8aᶠ/ᶠ; ᴼᵀ⁻ᴵ*) and LRRC8A KO (*Lrrc8aᶠ/ᶠ, Cd4-Cre; OT-I*) mice were activated by N4 peptide and its variants (Q4, V4, G4), at indicated concentration for 6 h ex vivo. Cells were gated on TCRβ⁺ CD8⁺ Vα2⁺. Nur77 expression (**a–d**), phosphorylated S6 and ERK1/2 (**e–i**), TNFα (**j**), CD71 and CD98 (**k, l**) were examined by flow cytometry. **m–p** Cell proliferation by OT-I peptides ex vivo were shown by CFSE dilution at 48 h after activation. Representative flow cytometry plots (left) and the quantification of percentage (right) were shown for all the panels (n = 3 replicates per mouse for all the panels). Data are representative of two (**a–d, g, h, j–l, n–p**), three (**e, f, i**), or four (**m**) independent experiments. Unpaired two-sided *t* test was used for all the panels. Data are presented as mean ± SEM. Source data are provided as a Source Data file.

impact of RVD via LRRC8A on T cell activation depends on the strength of the TCR signal.

To further test if TCR-pMHC affinity influences the role of LRRC8A in T cell activation, we exploited previously characterized OT-I peptide variants with lower TCR signaling potency compared with N4, SIIQ-FEKL (Q4), SIIVFEKL (V4), SIIGFEKL (G4) (in order of highest to lowest affinity)[39,40]. Intriguingly, stimulated with a series of OT-I peptide mutants, LRRC8A-deficient OT-I CD8⁺ T cells behaved less effectively activated (Fig. 2h). Like N4 cognate, less saturated doses of OT-I peptide mutants demonstrated more dramatic defects in LRRC8A KO cells. Moreover, more pronounced differences were observed by decreasing the peptide affinity (from Q4 to G4) (Fig. 2h). These data strongly support the requirement of cell volume regulation mediated by LRRC8A during T cell activation, and the weight of cell volume regulation by LRRC8A in T cell activation relies on TCR signal strength.

### T cell signaling and function are defective in LRRC8A KO T cells upon activation

T cell activation defects are usually due to dysregulated T cell signaling. To examine if LRRC8A controls TCR signaling during T cell blast, we first assessed the Nur77 expression, which reflects the immediate-early response of TCR stimulation. The induction of Nur77 in LRRC8A KO OT-I CD8⁺ T cells was significantly attenuated stimulated by OT-I peptides with different affinities (Fig. 3a–d). Concomitantly, phosphorylation of S6 (p-S6^S240/244) and MAPK (p-ERK1/2^T202/Y204), signaling downstream of TCR activation, was compromised (Fig. 3e–i). The hindered TCR signaling existed in antigen-independent activation by anti-CD3ε antibody stimulation in both CD4⁺ and CD8⁺ T cells as well (Supplementary Fig. 4a–c). In line with the TCR signaling deficiency in the absence of LRRC8A, T cell function, such as cytokine production, was curtailed upon T cell activation based on intracellular staining of TNFα (Fig. 3j and Supplementary Fig. 4d). In addition, the nutrient markers CD71 and CD98 induction were impeded in LRRC8A-deficient T cells upon TCR activation (Fig. 3k, l). Furthermore, LRRC8A-deficient T cells proliferated less upon TCR activation by OT-I peptides and anti-CD3ε antibodies, shown by CFSE dilution after cell division (Fig. 3m–p, and Supplementary Fig. 4e). Moreover, in previously activated CD8⁺ T cells, restimulation of LRRC8A KO cells produced significantly less TNFα and IFNγ (Supplementary Fig. 4f). Lastly, ectopic expression of full-length LRRC8A (*Lrrc8aᶠᴸ*) in primary splenic LRRC8A KO CD8⁺ T cells restored their protein expression and robustly enhanced T cell function, shown by cytokine production after restimulation (Supplementary Fig. 4g, h). Collectively, in support of T cell activation deficit in the absence of LRRC8A, T cells displayed inadequate TCR signal, compromised proliferation and function without LRRC8A. Similar results were obtained with the lower affinity N4 variants (Q4/V4/G4) for the CD71 and CD98 expression (Supplementary Fig. 5a–f), phosphorylation of ERK1/2 (Supplementary Fig. 5g–i), and TNFα production (Supplementary Fig. 5j–l).

### Distinct transcriptional profiles relying on TCR stimulation strength require LRRC8A-mediated cell volume regulation

To assess whether cell volume regulation mediated by LRRC8A modulated distinctive T cell transcriptional profiles, we activated OT-I CD8⁺ T cells using N4 peptide at high (N4^high, NH, 1000 pM) and low (N4^low, NL, 10 pM) doses, and G4 peptide (1000 nM) based on results from Fig. 2. Splenic OT-I CD8⁺ T cells were activated in vitro for 6 h and sorted for RNA sequencing (RNA-seq). TCR signal strength order of

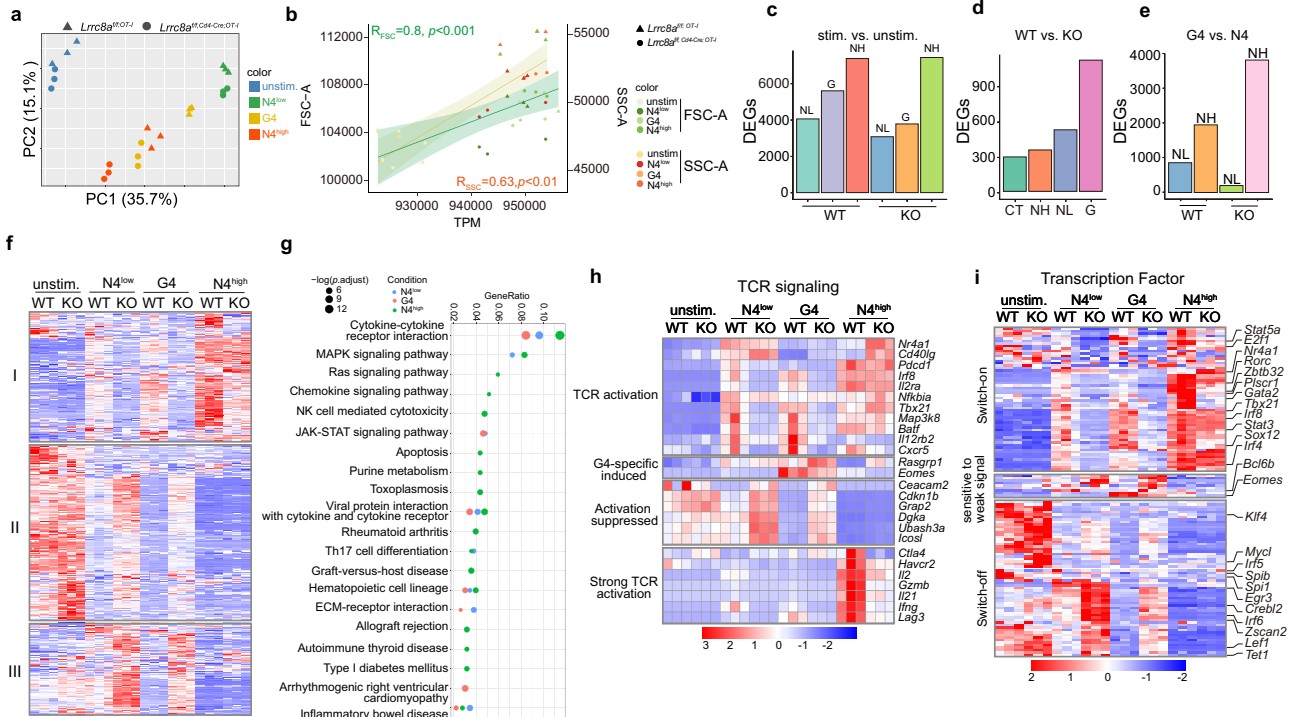

**Fig. 4 | Transcriptional profiles relying on TCR stimulation strength require LRRC8A-mediated cell volume regulation.** Transcriptomic analysis of RNA samples of OT-I CD8$^+$ T cells from WT (*Lrrc8a$^{f/f}$*) and KO (*Lrrc8a$^{f/f, Cd4-Cre}$*) mice, stimulated with OT-I peptides at indicated concentration for 6 h ex vivo. N4$^{low}$ or NL (10 pM), G4 (1000 nM), N4$^{high}$ or NH (1000 pM). n = 3 mice per group. **a** Principal component analysis (PCA) of the normalized expression data demonstrated distinct clusters. **b** The correlation analysis between FSC/SSC and RNA reads by sequencing (total transcripts per million, TPM). The error bands were mean ± SD. **c**–**e** Differentially expressed genes (DEGs) identified between different groups. Stimulation (stim.)

(NL, G, NH) groups versus unstimulated (unstim.) group (**c**), WT versus KO in each treatment (**d**), and G4 versus N4 (**e**). **f** Heatmap of DEGs across all the groups. I, genes switched on by TCR signal; II, genes switched off by TCR signal; III, genes sensitive to LRRC8A-deficiency. **g** KEGG pathway analysis of DEGs between WT (*Lrrc8a$^{f/f; OT-I}$*) versus KO (*Lrrc8a$^{f/f; Cd4-Cre; OT-I}$*) under various stimulations. **h**–**i** Heatmap of DEGs associated with TCR signaling pathways (**h**) and transcription factors (**i**). One-way ANOVA test was performed to evaluate the statistical significance in (**b**). Source data are provided as a Source Data file.

OT-I peptide variants was N4$^{high}$ > G4 > N4$^{low}$ (Supplementary Fig. 6a). Principal component analysis (PCA) demonstrated well-separated clusters for different treatments (Fig. 4a). Within each treatment, two groups were distinctive based on the genotypes. To determine the relationship between cell size and anabolism driven by TCR signals, we first analyzed the total number of gene transcripts detected by RNA-seq. A positive correlation existed between the mRNA transcription activity (transcripts per million cells, TPM) and FSC and SSC (Fig. 4b). This indicated that FSC and SSC could reflect the T cell anabolism triggered by different strengths of TCR signals. Further analysis between stimulated and unstimulated groups showed an increased number of gene expressions modulated in concert with TCR signal strength (NH > G > NL) (Fig. 4c). Moreover, considerably more genes were differentially expressed (DEGs) in lower TCR signals between WT and KO groups, particularly in G4 and NL treatments (Fig. 4d, e). This supports our previous results that LRRC8A had a more remarkable impact on T cell activation, especially for unsaturated TCR signals (Fig. 2b, h). We then focused on the transcriptomic analysis between high and low TCR signal strength groups. Heatmap of DEGs across different groups revealed distinct transcriptional profiles relying on TCR signal strength (Fig. 4f). There are at least three different patterns of gene expression: (I) induced by the TCR signal (switch-on); (II) repressed by the TCR signal (switch-off); (III) more induced in LRRC8A KO cells with lower TCR signal (Fig. 4f). TCR signaling initiated group I genes' expression relying on the signal strength, preferentially to the stronger one. In addition, the TCR signal elicited by the same concentration of OT-I peptides could initiate group I gene expression, but to a lesser extent in LRRC8A KO cells, suggesting group I genes

transcription demands sufficient TCR signal strength endowed by LRRC8A. On the contrary, the low TCR signal suppressed a plethora of gene expression (group II), mainly in WT OT-I CD8$^+$ T cells but not LRRC8A-deficient cells, implying TCR signal strength regulated by LRRC8A confers the level of shutdown of their transcription (Fig. 4f). Most intriguingly, group III genes were only upregulated in LRRC8A KO cells under lower TCR signal strength (N4$^{low}$ and G4), but not strong TCR signal, displaying a trend completely different from group I and II (Fig. 4f). Further GO enrichment analysis for the group III genes revealed that the GTPase activity, guanyl nucleotide binding and guanyl ribonucleotide binding pathways are top enriched (Supplementary Fig. 6b).

To understand the biological processes that LRRC8A implicates, we performed the Kyoto encyclopedia of genes and genomes (KEGG) pathway analysis (Fig. 4g). Enriched KEGG pathways varied in different TCR signal strength groups (Fig. 4g). Significant pathways enrichment in high dose of N4 treatment were cytokine-cytokine receptor interaction, MAPK, RAS, and Th17 cell differentiation. Meanwhile, G4 and NL groups showed similar pathway enrichment, including cytokine-cytokine receptor interaction, JAK-STAT signaling, viral protein interaction with cytokine and cytokine receptors, hematopoietic cell lineage and ECM-receptor interaction (Fig. 4g). It is worth noticing that cytokine-cytokine receptor interaction was top enriched in all three treatments. Furthermore, most cytokines and chemokines, as well as genes involved in the biosynthesis of amino acids, were induced by TCR signals while stronger in WT cells compared with LRRC8A-deficient cells, suggesting more robust anabolism in WT cells (Supplementary Fig. 6c).

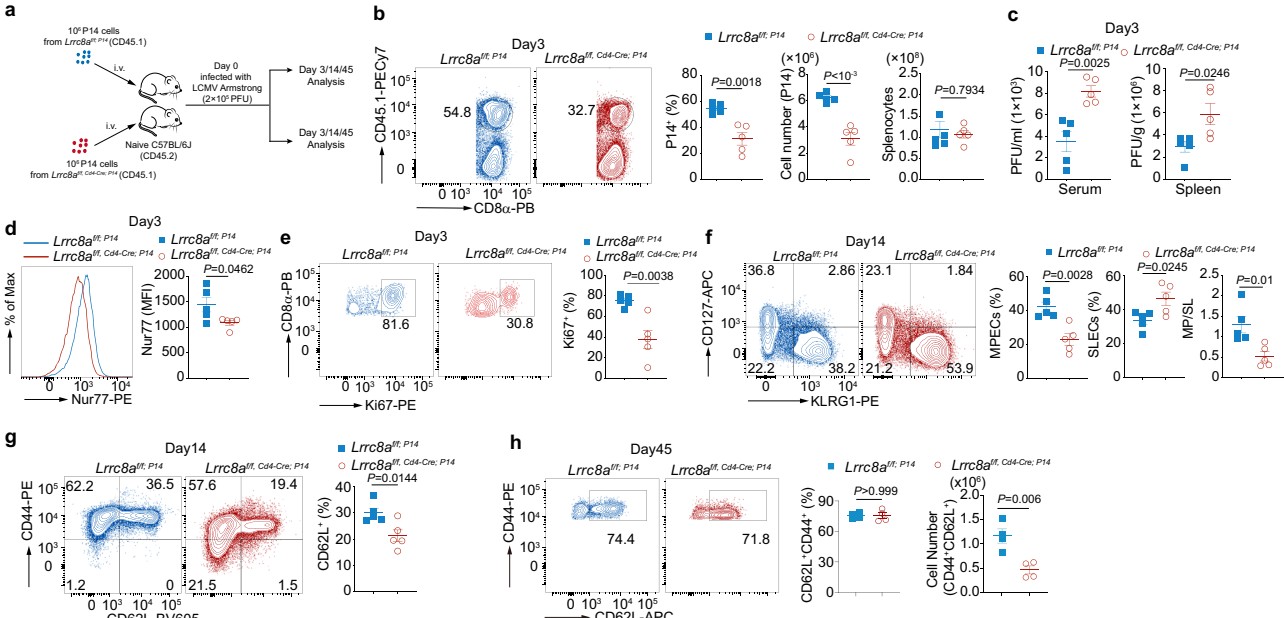

**Fig. 5 | LRRC8A deficiency impairs T cell-mediated antiviral immunity to LCMV Armstrong infection. a** Experimental design of LCMV Armstrong infection (2 × 10⁵ PFU). 10⁶ P14 cells isolated from WT (*Lrrc8a^{f/f; P14}*) or KO (*Lrrc8a^{f/f, Cd4-Cre; P14}*) mice (CD45.1⁺) were adoptively transferred to B6 mice (CD45.2⁺). **b** P14 T cells (Vα2⁺CD45.1⁺CD8⁺) in the spleen on the day 3 pi (*n* = 5 mice/group). **c** Virus load detected in the serum and spleen from mice in (**b**), measured by RT-qPCR (*n* = 5 mice/group). **d** Expression of Nur77 in P14⁺ T cells in spleen from mice in (**b**) (*n* = 5 mice/group). **e** Ki67 expression in P14 cells in the spleen from mice in (**b**) (*n* = 5 mice/group). **f–g** Evaluation of CD8⁺T cell effector/memory status. (**f**) CD127 and

KLRG1 staining of P14 CD8⁺ T cells from mice in (**a**) on the day 14 pi. MPECs (CD127⁺KLRG1⁻) and SLECs (CD127⁻KLRG1⁺) (*n* = 5 mice/group). **g** CD62L^{hi} memory T cells from mice in (**f**) (CD45.1⁺CD8⁺Vα2⁺CD44⁺CD62L⁺) on day 14 pi (*n* = 5 mice/group). **h** CD62L^{hi} memory T cells (CD45.1⁺CD8⁺Vα2⁺CD44⁺CD62L⁺) from mice in (**a**) on day 45 pi (*n* = 4 mice/group). Representative flow cytometry plots were shown on the right and their quantification on the left for (**b**, **d–h**). Data are representative of two (**c**, **g**, **h**) or three (**b**, **d–f**) independent experiments. Two-sided unpaired *t* test was used in (**b–h**), Data are presented as mean ± SEM. Source data are provided as a Source Data file.

TCR signaling-related molecules were further analyzed. Based on previous findings, we nominated four modules of genes that were dose- and LRRC8A-dependent transcriptional activation or suppression: (1) TCR activation induced module, including *Nr4a1*, *Cd40lg*, *Pdcd1*, *Il2ra*, *Tbx21*, and *Cxcr5*. (2) G4-specific module, including *Rasgrp1* and *Eomes*. (3) activation suppressed module, which tends to be suppressed by activating TCR signal, remarkably with more robust TCR strength. Genes included were *Ceacam2*, *Cdkn1b*, *Grap2*, *Dgka*, *Ubash3a*, and *Icosl*. (4) Strong TCR signal activation module. This module can only be induced at N4^{high} group, to a less extent in LRRC8A KO cells, including *Ctla4*, *Havcr2*, *Il2*, *Gzmb*, *Il21*, *Ifng*, and *Lag3*. Remarkably, Th1 and Th17 cytokines such as *Il2*, *Ifng*, and *Il21* could only be strongly induced in WT cells. Moreover, inhibitory receptors like *Ctla4*, *Havcr2*, and *Lag3* also required potent TCR signals to get induced, contrary to *Pdcd1* (Fig. 4h).

One of the most magnificent events upon T cell activation is the initiation of substantial mRNA transcription, observed in our RNA-seq results. mRNA transcription is primarily due to the transcription factors (TFs), and subsequently determines the T cell fate[41]. In line with Fig. 4f, differentially expressed TFs exhibited similar patterns. At least three categories of TFs should be mentioned (Fig. 4i). (1) "switch-on" motif by TCR signaling, positively correlating with TCR signal strength, including *Stat5*, *E2f1*, *Nr4a1*, *Rorc*, *Zbtb3*, *Gata2*, *Tbx21*, *Irf8*, *Stat3*, *Sox12*, and *Irf4*. (2) weak TCR signal-sensitive motif in N4^{low} and G4 groups, better upregulated even in LRRC8A KO cells, *e.g.*, *Bcl6b* and *Eomes*. (3) "switch-off" motif by TCR signaling, negatively correlating with TCR signal strength, containing *Klf4*, *Mycl*, *Spi1*, *Egr3*, *Irf5*, *Irf6*, *Lef1*, and *Tet1*. Based on TFs expression profiles, it is rational to postulate that LRRC8A participates in CD8⁺ T cell effector and memory function. Together, our results revealed distinctive transcriptional profiles in WT and LRRC8A KO T cells in response to various TCR activating signals,

suggesting LRRC8A plays a prominent role in T cell activation, differentiation, effector, and memory function.

## LRRC8A deficiency attenuates CD8⁺ T cell-mediated antiviral immunity to LCMV Armstrong

In light of previous ex vivo results and transcriptome analysis, it is reasonable to hypothesize that the ablation of LRRC8A in T cells leads to functional defects in vivo. To investigate whether LRRC8A regulates the CD8⁺ T cell-mediated antiviral immunity in vivo, we adoptively transferred the control (*Lrrc8a^{f/f; P14}*) and LRRC8A KO (*Lrrc8a^{f/f, Cd4-Cre; P14}*) CD8⁺ T cells in the P14 TCR transgenic background into B6 wild-type mice, and then infected with lymphocytic choriomeningitis virus (LCMV) strain Armstrong intraperitoneally (Fig. 5a). Recipient mice were euthanized on days 3 and 14 post-infection, and their antiviral immunity was evaluated. We recovered the same amount of splenocytes in WT and KO mice, but control P14 CD8⁺ T cells were significantly more expanded than LRRC8A KO cells, both in percentage and absolute number (Fig. 5b). As a result, recipients with LRRC8A KO cells bore significantly higher virus load in serum and spleen, suggesting its defective antiviral ability at the early stage of this acute infection (Fig. 5c). In addition, T cell activation marker Nur77 is higher in mice with control P14 cells, reflecting stronger T cell signal and activation (Fig. 5d). In concert with it, LCMV infection elicited a more robust proliferation of control cells than LRRC8A-deficient P14 CD8⁺ T cells based on the expression of the proliferation marker Ki67 (Fig. 5e). Meanwhile, all the WT and LRRC8A KO P14 CD8⁺ T cells experienced the substantial amplification after LCMV infection by day 3, shown by diluted CellTrace Violet labeling (Supplementary Fig. 7a). The CD44 expression was the same as well (Supplementary Fig. 7b).

LCMV strain Armstrong causes an acute infection where the virus is eliminated about 7–8 days post-infection[42]. In response to acute

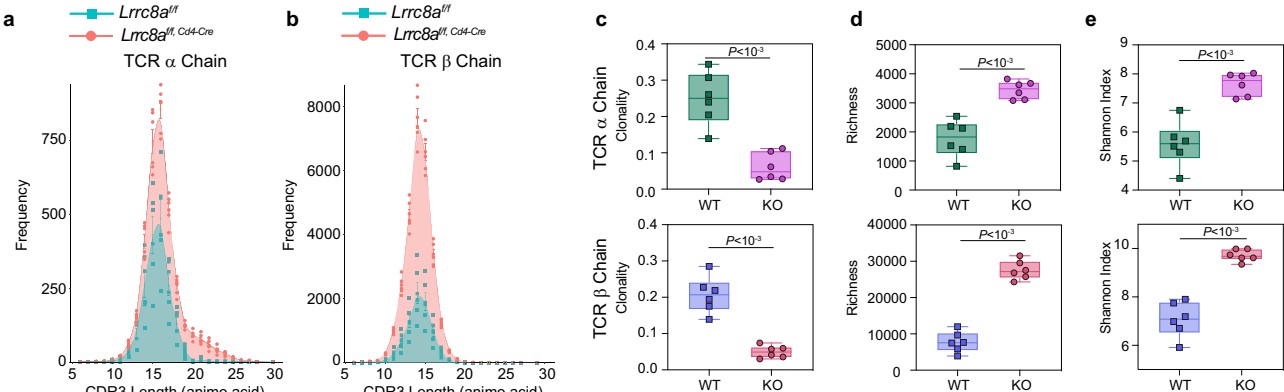

**Fig. 6 | LRRC8A shapes the TCR repertoire in the thymus.** TCR-seq analysis of thymocytes from WT (*Lrrc8a^f/f*) and KO (*Lrrc8a^f/f, Cd4-Cre*) mice (*n* = 6 mice/group). Age- and sex-matched littermates were used for sequencing (see details in methods). **a, b**, Complementary determination region 3 (CDR3) distribution of TCR α and β chain. **c–e** Box-and-whisker plots of the clonality (**c**) richness (**d**) and Shannon diversity index (**e**) for TCR α and β chain with boxes marking first quartile, median, and third quartile and with whiskers extending 1.5 times the inter-quartile range. Each point represents one mouse (*n* = 6 mice/group). Data are representative of two independent experiments. Unpaired two-sided *t* test was used in (**c–e**). Data are presented as mean ± SEM. Source data are provided as a Source Data file.

virus infection, P14 CD8+ T cells are activated, expanded, and differentiated into effector cells to eradicate the virus. After the virus clearance, the expanded antigen-specific CD8+ T cells differentiate into two major subsets of effector cells, terminally differentiated short-lived effector cells (SLECs, KLRG1^hi CD127^lo) and memory precursor effector cells (MPECs, KLRG1^lo CD127^hi) which can form long-lived memory CD8+ T cells eventually[43]. To test if LRRC8A has a role in T cell memory establishment after the virus resolved, we examined the P14 CD8+ T cells at day 14 pi. Virus were undetectable at day 14 pi (Supplementary Fig. 7c). Recipient mice transferred with LRRC8A KO P14 cells have more SLECs and fewer MPECs, implying insufficient generation of MPECs (Fig. 5f). In addition, fewer LRRC8A KO P14 cells differentiated into CD62L^hi memory-like T cells (Fig. 5g). The CD44^low subset had undergone proliferation after viral infection by day 14 as well, consistent with cell division observed at day 3 pi (Supplementary Fig. 7d). Moreover, at day 45 pi, memory T cell numbers were significantly lower in LRRC8A KO recipients compared with WT controls (Fig. 5h). Together these data demonstrated that LRRC8A is demanded in T cell-mediated antiviral immunity, including the process of virus elimination and T cell memory establishment.

## LRRC8A shapes the TCR repertoire in the thymus

T cell development undergoes thymic selections, which tightly depend on TCR signaling strength triggered by self-antigen presented via MHC[44,45]. To examine whether the TCR signaling controlled by LRRC8A contributes to the formation of TCR repertoire, we performed the TCR sequencing (TCR-seq) analysis of thymocytes from WT (*Lrrc8a^f/f*) and LRRC8A KO (*Lrrc8a^f/f, Cd4-Cre*) mice. Complementary determination region 3 (CDR3) sequences unambiguously demonstrated highly similar distribution patterns in length for both TCR α and β chains (Fig. 6a, b). Interestingly the clonality in the thymus of LRRC8A KO mice was decreased (Fig. 6c). The clonality was assumed to reflect the proliferation of T cells that get through the thymic positive selection. In line with it, the decrease of clonality might be due to the compromised T cell signaling strength with the loss of LRRC8A. Also, the richness of CDR3 in the thymus of LRRC8A KO mice was higher, indicating a greater number of unique CDR3 sequences obtained (Fig. 6d). Consequently, the thymus of LRRC8A KO mice had a significantly higher Shannon diversity index (see details in materials and methods) both in TCR α and β chains (Fig. 6e). Moreover, VJ recombination in both TCR α and β chains displayed extraordinarily similar (Supplementary Fig. 8a, b), suggesting non-deviation in pairing VJ segments during T cell developments. Thus, the TCR repertoire in mice without

LRRC8A is more even and diverse. Given that TCR signaling is impeded in the results mentioned above, it is plausible that the activation and proliferation of T cells without LRRC8A through thymic selection have been compromised, which consequently shapes TCR repertoire in the thymus.

## Cell volume regulation via RVD mediated by LRRC8A fine-tunes TCR proximal signal by maintaining molecular density during T cell blast

In principle, T cell enlargement by the cumulative anabolic biosynthesis during activation is equivalent to the hypotonic cell swelling in terms of temporal slow-extreme, in which the VRAC formed by LRRC8 family proteins regulates the RVD. To reveal the mechanisms of cell volume regulated by LRRC8A during T cell activation, we attempted to simulate the gradually slow cell size increase during T cell activation by acute hypotonic cell swelling. We evaluated TCR stimulation-induced acute T cell activation and TCR proximal signal under isotonic (300 mOsm) or hypotonic conditions (200 mOsm). In order to minimize the adverse effect of the low tonicity environment on T cell vitality and responsiveness, we kept T cells alive in the moderate hypotonic buffer during stimulation, and there was no significant cell death difference between WT and LRRC8A deficient T cells in our experimental settings (Supplementary Fig. 9a). We observed attenuated CD25 induction upon TCR stimulation in WT OT-I CD8+ T cells under hypotonicity compared with isotonicity (ΔCD25=CD25_{Iso}-CD25_{Hypo}). At the same time, LRRC8A deficiency resulted in a more pronounced loss of CD25 induction due to the failure of cell volume control (RVD) in hypotonic cell swelling (Fig. 7a). Similarly, by cross-linking CD3ε, CD25 expression in LRRC8A KO cells dropped more in a hypotonic environment, both for splenic CD4+ and CD8+ T cells (Supplementary Fig. 9b, c). Besides CD25, upon T cell activation, CD71 and CD98 were less induced in LRRC8A KO in response to hypotonicity than WT cells (Fig. 7b and Supplementary Fig. 9d, e).

Considering LRRC8A-deficient cells inflated much larger due to the lack of RVD under hypotonic conditions, we logically postulated the attenuated TCR signal by the decreased molecular density of TCR signaling machinery (Supplementary Fig. 9f). To test this hypothesis, we first examined the expression level of the key TCR signaling molecules, such as LCK, FYN, ZAP70, LAT, and T cell membrane receptors, including CD4, CD8, CD45, CD3ε, CD28, TCRβ, in both isotonic and hypotonic conditions. The expression of these molecules was unaltered when WT and LRRC8A KO cells were immersed in a hypotonic buffer (Fig. 7c and Supplementary Fig. 9g). Meanwhile,

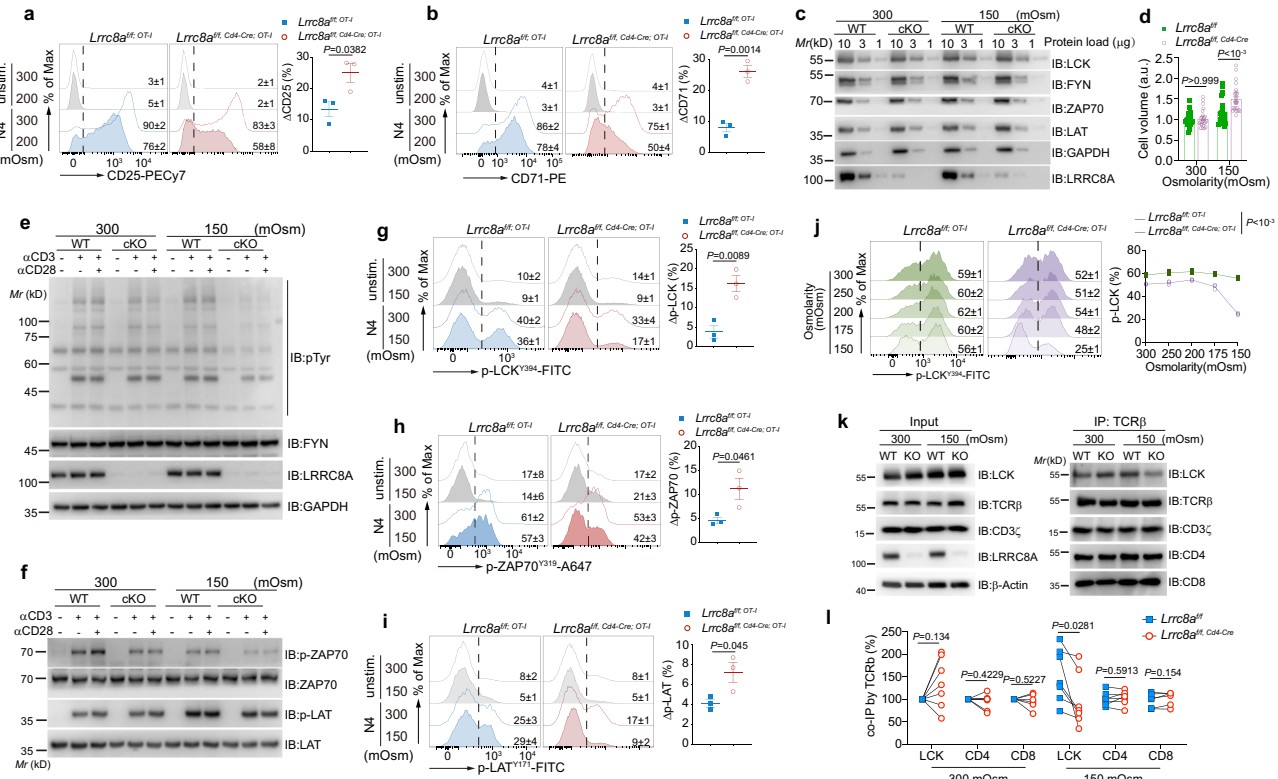

**Fig. 7 | Cell volume regulated by LRRC8A during T cell activation fine-tunes TCR proximal signal by keeping molecular density.** T cells were stimulated by OT-I peptide or antibodies in isotonic or hypotonic buffer at indicated conditions. **a, b** Expression of CD25 (**a**) and CD71 (**b**) measured by flow cytometry. Splenic OT-I CD8$^+$ T cells from *Lrrc8a$^{f/f; OT-I}$* or *Lrrc8a$^{f/f, Cd4-Cre; OT-I}$* mice were activated by N4 peptide (1 nM) ex vivo for 6 h under isotonic or hypotonic condition and gated on TCRβ$^+$CD8$^+$Vα2$^+$ (n = 3 replicates/mouse). **c** Immunoblots of TCR proximal signaling molecules in thymocytes from WT (*Lrrc8a$^{f/f}$*) or cKO (*Lrrc8a$^{f/f, Cd4-Cre}$*) mice, in isotonic (300 mOsm) and hypotonic (150 mOsm) solutions. **d** Cell volume of thymocytes from *Lrrc8a$^{f/f}$* or *Lrrc8a$^{f/f, Cd4-Cre}$* mice under isotonic (300 mOsm) and hypotonic (150 mOsm) condition for 15 min. n = 23–26 cells/group. **e–f** Immunoblots of p-Tyr (**e**), p-ZAP70$^{Y319}$, and p-LAT$^{Y171}$ (**f**) under isotonic (300 mOsm) and hypotonic (150 mOsm) from *Lrrc8a$^{f/f}$* or *Lrrc8a$^{f/f, Cd4-Cre}$* mice stimulated or not with anti-CD3ε (1 μg/ml) or anti-CD3ε (1 μg/ml) and anti-CD28 (1 μg/ml) for 2 min at 37 °C.

FYN and GAPDH were used as loading control. **g–i** Flow cytometry of p-LCK$^{Y394}$ (**g**), p-ZAP70$^{Y319}$ (**h**) and p-LAT$^{Y171}$ (**i**). The OT-I T cells were stimulated with N4 (1 nM) in the indicated solutions for 30 min. n = 3 replicates/mouse for (**g–i**). **j** p-LCK$^{Y394}$ in OT-I CD8$^+$ T cells stimulated as in (**g**) under the osmolarity ranging from 300 to 150 mOsm (n = 3 replicates/mouse). **k, l** Immunoprecipitation of TCRβ and followed by immunoblot of key molecules in TCR complex in the thymocytes isolated from *Lrrc8a$^{f/f}$* and *Lrrc8a$^{f/f, Cd4-Cre}$* mice. Representative blot in (**k**) and quantification in (**l**) (n = 7 mice/group). Representative flow cytometry histogram (left) and the quantification (right) were shown for panels (**a, b, g–j**). Data are representative of two (**c–f, h**), three (**a, b, g, i**), or seven (**k**) independent experiments. Calculation was done by the formulas: ΔX (%) = X$_{Iso}$-X$_{Hypo}$ (%). Unpaired two-sided *t* test was used in **a, b, g–i**. Two-way ANOVA with Bonferroni's post-hoc test was in (**d, j**). Two-sided paired *t* test was used in (**l**). Data are presented as mean ± SEM. Source data are provided as a Source Data file.

osmotic swelling drove a significant cell size increase in LRRC8A KO cells compared to WT cells (Fig. 7d). Akin to the easy-to-obtain sufficient thymocyte cell amount and their relatively small cytoplasm, we subsequently used these cells to evaluate the TCR activating signal. Thymocytes were stimulated by crosslinking anti-CD3ε (1 μg/ml) and anti-CD28 (1 μg/ml) for 2 min. Under this condition, cells are not blasted within minutes, even with a strong TCR activating signal. In the meantime, we observed a similar level of total tyrosine phosphorylation between WT and LRRC8A KO cells in isotonic buffer, indicating LRRC8A is not directly involved in the control of TCR signaling before cell blast (Fig. 7e). However, TCR signaling was severely compromised in LRRC8A-deficient cells under hypotonic conditions (Fig. 7e). Given that the TCR molecule expression was unaltered, but cell volume was enlarged in acute hypotonic swelling, more giant cells failed to be efficiently activated, presumably due to the loss of appropriate TCR signal molecular density. In agreement with it, two primary components of proximal TCR machinery, ZAP70 and LAT were less phosphorylated (p-ZAP70 and p-LAT) in LRRC8A KO cells under hypoosmotic swelling (Fig. 7f). Consequently, p-AKT and p-ERK were mitigated in LRRC8A-deficient cells in the hypotonic buffer in response to T cell activation (Supplementary Fig. 9h). These data strongly indicate that the compromised T cell activation in hypotonic conditions,

especially for LRRC8A KO cells, is probably owing to the excessive drop of molecular density.

Furthermore, we verified pMHC and anti-CD3ε crosslinking-elicited TCR proximal signal at single cell level in swollen cells. Splenic T cells were activated by either N4 peptide (1 nM) or anti-CD3ε (1 μg/ml) in the isotonic or hypotonic (150 mOsm) solution for 30 min, followed by phospho-flow cytometry. Proximal TCR signals in WT cells were subtly altered, including phosphorylated LCK, ZAP70, and LAT (p-LCK$^{Y394}$, p-ZAP70$^{Y319}$, and p-LAT$^{Y171}$), whereas they were more markedly compromised in LRRC8A KO T cells when cells were immersed in hypotonic buffer (Fig. 7g–i and Supplementary Fig. 9i). Moreover, p-LCK$^{Y394}$ gradually decreased in LRRC8A-deficient T cells when the tonicity was lowered from 300 mOsm to 150 mOsm, with a sharp drop at 150 mOsm (Fig. 7j). In the meantime, p-LCK$^{Y394}$ in WT cells remained almost constant (Fig. 7j). Of note, the viability of CD8$^+$ T cells from both WT and LRRC8A KO mice in the aforementioned manipulations was the same, with decreased osmolarity during stimulation (Supplementary Fig. 9j, k). Moreover, to exclude the possible involvement of nutrients in our experimental settings, we first used PBS to dilute RPMI. Reducing half the amount of nutrients in the buffer without altering the osmolarity did not significantly modulate TCR signaling, as shown by p-LCK (Supplementary Fig. 9l, m). On the

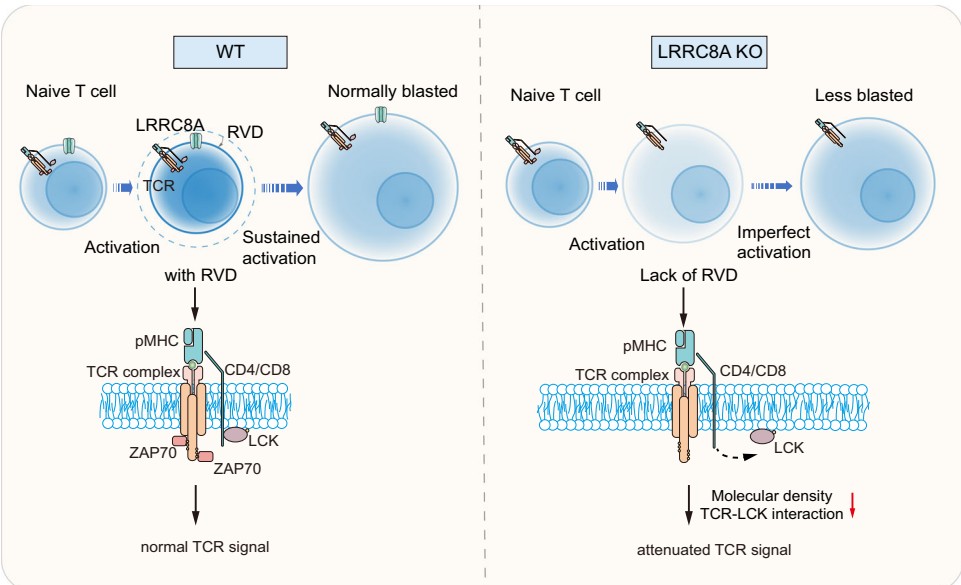

**Fig. 8 | Proposed model for the control of T cell activation by LRRC8A-formed VRACs.** Naïve T cells begin to blast upon TCR activation, and RVD mediated by LRRC8A-formed VRACs limits the cell size. As a result, the TCR signaling molecules are maintained at an appropriate density for optimal T cell activation and blast. In the absence of LRRC8A, T cells lose their regulation of cell size during blast, and TCR signal decays due to the reduced density of signaling molecules. Eventually T cells fail to be adequately activated.

other hand, using mannitol to adjust the buffer tonicity[28], we also observed a more pronounced decrease of p-LCK in LRRC8A KO T cells upon TCR activation (Supplementary Fig. 9n, o). Thus, it is clear that nutrients have barely effects in our settings.

Considering the unaltered proximal TCR signaling molecules expression level for WT and LRRC8A KO cells during acute hypotonic treatment, it is rational to hypothesize that the TCR signal defectiveness is due to the decrease of molecular number density or spatial conformation change in enlarged LRRC8A KO cells. To assess it, we immunoprecipitated TCRβ and blot the key TCR complex molecules. CD3ζ, CD4, and CD8 remain attached to TCRβ. However, LCK exhibited significantly less association with TCR complex in the absence of LRRC8A in hypotonic cell swelling (Fig. 7k, l). Direct interaction between TCRβ and LRRC8A was not observed neither before nor after stimulation (Supplementary Fig. 9p). Taken together, our data provided undoubted evidence that cell volume regulation mediated by LRRC8A controls TCR signaling by keeping molecular density at a relatively constant level instead of an excessive decline of density, mitigating the signal transduction during T cell blast (Fig. 8).

## Discussion

Here we revealed the fundamental role of cell volume augmentation contributing in T-cell activation and T-cell-mediated antiviral immunity. Firstly LRRC8A-formed VRAC is obligatory for optimal T cell activation and function. Pharmacological inhibition of VRAC or genetic ablation of VRAC formed by LRRC8A in T cells causes defective activation, proliferation, and function regardless of antigen-specific TCR signal or not. Notably, the impact of LRRC8A on T cell activation is more evident for the unsaturated TCR signal and low affinity cognate, suggesting LRRC8A is implicated in setting the optimal TCR activating signaling threshold, presumably via the RVD process. Second, distinct transcriptional profiles relying on TCR stimulation strength require LRRC8A-mediated cell volume regulation. Third, the deletion of LRRC8A in T cells attenuates antiviral immunity in vivo. Moreover, under physiological conditions, LRRC8A participates in shaping TCR repertoire during thymic selection. Finally, utilizing a hypotonic buffer to simulate the osmotic swelling driven by anabolism during T cell blast, we proved that the proximal signal defects in LRRC8A-deficient

cells were due to the excessively decreased molecular density in the TCR complex as a consequence of the cell volume enlargement (Fig. 8).

Lymphocyte blast during activation is well-known, but its physiological role and underlying mechanisms are poorly understood. Multiple pathways are involved in regulating cell size under homeostasis, including PI3K/AKT/mTORC1 and Hippo/YAP1[5,6]. Moreover, cell volume is mainly determined by intracellular osmotic contents and extracellular osmolarity. One of eukaryotes' most fundamental cellular functions is to constantly adapt in response to osmotic shift between the intra- and extracellular osmolarity to avoid cell lysis by swelling[1]. Given that the body fluid osmolarity remains almost constant under physiological conditions, the perturbation of cell volume is usually due to intracellular osmolarity. In the scenario of lymphocytes, particularly T cells, substantial anabolic biosynthesis initiated by the activating TCR signal drives robust cumulation of osmotic active molecules intracellularly, leading to the water influx and resulting in T cell volume increase in concert with the transition from naïve to effector cells eventually[13,14]. mTOR activity in CD4+ T cells correlates to cell size during T cell activation, reflecting the cell fate determination by intracellular metabolic programs[46]. Unlike cell size regulation in steady state, the RVD process functions after osmotic swelling, implying LRRC8A-formed VRAC controls T cell signal during T cell swelling after activation, but not before. In this study, our data provided clear evidence to uncover the importance of appropriate cell volume control in T cell activation and function during T cell blast, owing to the preservation of sufficient molecular density via the RVD process mediated by LRRC8A.

Besides the canonical VRAC function of LRRC8 family proteins, recent studies reported LRRC8-formed channels could be permeable to other molecules, such as amino acids, taurine, and chemical drugs, as well as 2′3′cGMP-AMP (cGAMP), the agonist of STING pathway. However, the role of LRRC8A-formed VRAC in T cells remains controversial. Germline deletion of *Lrrc8a* in mice results in a dramatic loss of thymocytes and impaired B cell development[30]. However, a further study argued that the VRAC function is dispensable for T cell development in mice[31]. Recently, LRRC8C was reported to suppress CD4+ T cell function via cGAMP-STING-p53 signaling[32]. Upon α-CD3ε and α-CD28 stimulation, LRRC8C-deficient CD4+ T cells displayed decreased

p53 expression, which leads to more vigorous T cell proliferation and cytokine production[32]. Consistently, CD4+ T cell-dependent immunity, such as EAE and antiviral immunity to influenza, was compromised.

To solve this discrepancy, beneficial from T cell-specific KO mice (*Cd4-Cre*), we demonstrated that loss of LRRC8A did not affect T subsets at least in the thymus and spleen. Genetic ablation of LRRC8A recapitulated the pharmacological inhibition of VRAC effect on T cell activation, proliferation, and function, both in CD4+ and CD8+ T cells, regardless of antigen-specific or not. Cell volume regulation via RVD mediated by LRRC8A-formed VRAC has an evident impact on T cell activation, especially on weaker TCR signals. Intense or saturated TCR stimulation may overwhelm and cover the feedback effects of cell size on TCR signaling. Transcriptome analysis identified specific genes expression requiring LRRC8A during T cell activation, which may help to calibrate the TCR signaling threshold. In support of it, LRRC8A-deficient cells exhibited compromised anti-virus immunity in acute LCMV Armstrong infection. Surprisingly, under homeostatic conditions, TCR repertoire is affected mainly by LRRC8A during thymic selection, indicating suitable cell volume regulation aids the thymocytes in setting the threshold for activation and proliferation in the thymus. However, our data indicated no preference for TCR recombination by LRRC8A.

TCR signal triggering, propagation, and maintenance comprising a series of biophysical and biochemical reactions, is central to determining T cell fate and function[47,48]. Fully functional activation of T cells requires optimal kinetics and strength of the TCR stimulation[49–51]. Efficient activation of TCR signal and propagation demands appropriate spatial conformation of TCR signaling machinery[52–55]. Spatial-temporal dynamics of TCR signalosome conformation, including distance, distribution, and direction, guarantee the proceedings of TCR signals[55–58]. Our data revealed reciprocal interaction between TCR signaling and cell volume regulation during blast driven by activation. The TCR signal initiates substantial anabolic biosynthesis, which in turn raises the intracellular osmolarity creating the osmotic potential, leading to water influx, followed by turning on LRRC8A-formed VRAC to mediate the RVD. The subsequent RVD is critical for maintaining TCR signal strength by holding the molecular density for the signaling machinery. As a result, the TCR signal remains above the threshold to fully activate the T cells. Loss of LRRC8A interrupts the feedback loops between cell volume regulation and TCR signaling maintenance. At least, LCK remains in proper interaction with TCR with the presence of LRRC8A. Cell volume regulation probably contributes to the TCR mechano-transduction by modulating the *trans-* (pMHC-TCR) and *cis-* (lateral TCR-CD3 complex with associated receptors and kinases) interactions as well[59].

Our data demonstrated that the cell volume regulation via the RVD process mediated by LRRC8A-formed VRAC contributes to optimal T cell activation and function during T cell blasting. The T lymphocyte blast during activation is not only an accompanying phenomenon but also under rigorous control to assure proper T cell immunity.

## Methods
### Mice
All mice used in this study were on a C57BL/6 J (B6) genetic background. Sex-matched mice at the age of 6–12 weeks were used for the experiments. The mice with conditional allele of *Lrrc8a* were generated as described in Supplemental figures. Briefly, exon3 of wild-type allele of *Lrrc8a* was flanked by loxP sites using CRISPR-Cas9 system. *Lrrc8a^f/+* mice were crossed with *Cd4-Cre* mice to get the T-cell specific knockout mice (*Lrrc8a^f/f, Cd4-Cre*). To get the TCR transgene expression, *Lrrc8a^f/f, Cd4-Cre* mice were crossed with OT-I, OT-II, and P14 TCR transgenic mice respectively. CD45.1, OT-I, OT-II, and P14 mice were obtained from the Jackson Laboratory. *Lrrc8a^f/f* littermates were used as controls with the same transgenic background of the corresponding LRRC8A KO mice. Mice were maintained in specific pathogen free (SPF) animal facility. All animal experiments were performed under protocols approved by the Animal Care and Use Committee of School of Basic Medicine, Tongji Medical College, Huazhong University of Science and Technology.

### Antibodies and peptides
Rabbit anti-mouse LRRC8A monoclonal antibody was developed and produced by Abcam (ab254389). Antibody against mouse CD69 (H1.2F3, 1:500), CD25 (PC61, 1:500), CD4 (GK1.5, 1:1000), CD8α (53-6.7, 1:1000), TCRβ (H57-597, 1:500), Vα2 (B20.1, 1:500), CD3ε (145-2C11, 1:500), CD45 (30-F11, 1:500), CD45.1 (A20, 1:500), CD98 (RL388, 1:500), CD71 (R17217, 1:500), TNFα (MP6-XT22, 1:500), IFNγ (XMG1.2, 1:500), Nur77 (12.14, 1:500), CD28 (37.51, 1:500), CD48 (HM48-1, 1:500), Thy1 (30-H12, 1:500), CD44 (IM7, 1:500), Ki-67(16A8, 1:500), KLRG1 (2F1, 1:500), CD127 (SB1199, 1:500), phosphorylated LCK^Y394 (A18002D, 1:100), phosphorylated LAT^Y171 (A20005D, 1:100) and matched isotypes were from BioLegend (San Diego, CA). Protein A/G PLUS-Agarose (sc-2003) was from Santa Cruz Biotechnology (Santa Cruz, CA). Anti-mouse CD5 (53-7.3, 1:500) and CD62L (MEL-14, 1:500) were obtained from BD Biosciences (San Jose, CA). Monoclonal antibody against phosphorylated ZAP70^Y319 (MA5-28069, 1:100) were from Invitrogen (San Diego, CA), phosphorylated ERK^T202/Y204 (9101, 1:100), phosphorylated S6^S240/244 (5364, 1:100), phosphorylated tyrosine (4G10, 1:1000) antibodies were from Cell Signaling Technology (Beverly, MA). Antibodies for western blot of LCK (12477, 1:100), LAT (11326, 1:100), FYN (66606, 1:100), GAPDH (10494, 1:100), ZAP70 (60200, 1:100), β-actin (66009, 1:100), CD3ζ(12837, 1:100) were from Proteintech (Wuhan, China). OVA peptides 257–264 and 323–339 were from Anaspec. N4 mutants and LCMV gp33 peptide were synthesized by Sangon Biotech (Shanghai).

### Immunization in vivo
To induce antigen-specific T cell activation in vivo, OT-I CD8+ T cells, OT-II CD4+ T cells, or P14 CD8+ T cells were isolated from spleen, labeled with CellTrace Violet, and then transferred to B6 recipients intravenously respectively. In 24 h, OT-I and OT-II recipient mice were immunized with ovalbumin (500 μg in 200 μl PBS) (Sigma, A5503), and P14 recipients were infected with LCMV Armstrong (2 × 10^5 PFU), intraperitoneally. After 2–3 days, splenocytes were obtained from immunized mice, and antigen-specific activation and cell volume were examined by flow cytometry. Antigen-specific T cells were recovered by CellTrace Violet labeling.

### T cell activation and expansion in vitro
T cells were activated in vitro by anti-CD3ε crosslinking, mitogen concanavalin A (Sigma, L7647), or peptide-MHC. Briefly, splenocytes were freshly isolated from mice using lympholyte (Tebu-Bio). In a 96-well U-bottom plate, one million of splenocytes were incubated in RPMI1640 medium with 10% fetal bovine serum, 100 U/ml penicillin/streptomycin, containing anti-CD3ε (145-2C11), ConA, or synthetic peptides at indicated concentration for 6 h in cell culture incubator. To generate previously activated T cells, purified splenic CD4+ or CD8+ T cells were first stimulated with plate-bound anti-CD3ε antibody (145-2C11, 1 μg/ml) and soluble anti-CD28 antibody (37–51, 1 μg/ml) for 2 days, and then amplified in medium with IL-2 (50 U/ml) for 3 days. Restimulation was performed with plate-bound anti-CD3ε for 5 h or crosslinking anti-CD3ε (1 μg/ml) with anti-hamster antibody for 2 min at 37 °C. Alternatively, fresh thymocytes were activated by crosslinking anti-CD3ε and anti-CD28 (1 μg/ml) for 2 min at 37 °C. Activated T cells were further examined by flow cytometry or western blot for proliferation, cytokine production, and TCR signaling.

### T cell proliferation and cytokine production
CellTrace CFSE or Violet (Invitrogen) (1 μM) were used to label T cells and monitor their proliferation in vitro. T cells were first activated by

various ways mentioned previously, followed by flow cytometry on BD Fortessa cell analyzer (San Jose, CA). GolgiStop was supplemented during T cell activation for cytokine staining. Cells were stained for the surface markers first on ice for 30 min before fixation and permeabilization (BD, 554714). Intracellular staining of IFNγ and TNFα of permeabilized cells was evaluated by flow cytometry. Results were analyzed using FlowJo v10.5.

### LCMV Armstrong infections

LCMV Armstrong was propagated and quantified as previously described. To evaluate the acute P14 CD8$^+$ T cell activation, $10^6$ LRRC8A WT and KO cells in P14 background were transferred to B6 mice intravenously. 24 h later, frozen LCMV Armstrong stocks were thawed on ice and diluted in PBS, and $2 \times 10^5$ PFU were injected intraperitoneally. Acute T cell activation and proliferation were evaluated at day 3 pi, while effector to memory transition was monitored at day 14 pi by flow cytometry.

### VRAC inhibition

VRAC blocker, DCPIB (APExBio, B6780), was utilized to transiently inhibit VRAC function during T cell activation in vitro. DCPIB (25 μM) was added during T cell activation without any cytotoxicity. T cell activation, cytokine production, proliferation, and signaling were then verified by flow cytometry with or without DCPIB.

### LRRC8A overexpression in primary T cells

To ectopic expression of LRRC8A in primary T cells, mouse *Lrrc8a* cDNA clone was obtained from OriGene (Cat. 177725). *Lrrc8a* was cloned into a retrovirus-based plasmid pMIG-eGFP (Addgene: 9044). Retrovirus expressing LRRC8A was packaged in 293 T cells with pCL-Eco (Addgene: 12371). Primary CD8$^+$ T cells were negatively purified and activated with plate-coated anti-CD3ε (2 μg/ml) and soluble anti-CD28 (1 μg/ml) for 24 h, followed by two rounds of retrovirus infection at day 2 and 3. Infected T cells were expanded in complete RPMI1640 medium containing IL-2 (50 U/ml). Cells were used for experiments between day 4 and 5. eGFP$^+$ T cells indicated the restoration of LRRC8A expression.

### Hypotonic cell swelling and TCR signaling measurements

Isotonic buffer (300 mOsm) contains 100% RPMI-1640 medium. Hypotonic buffer was prepared by mixing isotonic solution (300 mOsm) with water as the following ratio ($V_{iso}$:$V_{H2O}$): 250 mOsm (5:1), 200 mOsm (2:1), 175 mOsm (1.4:1), 150 mOsm (1:1). T cells were stimulated at indicated osmolarity with the activating methods mentioned above, and T cell signaling was examined by western blot or phospho-flow cytometry. For thymocytes and previously activated T cells, after antibody crosslinking, cells were immediately lysed in 2x TNE buffer (2% NP-40, 4 mM EDTA, 100 mM Tris-Cl, pH 8.0) supplemented with protease and phosphatase inhibitor cocktail (Cell Signaling, 5872). Protein samples were subsequently separated by 8% SDS-PAGE and transferred to the PVDF membrane, subjected to blotting of total tyrosine phosphorylation (4G10) and TCR proximal signaling molecules. Methods for immunoblots were described previously (Wu, 2016, 2020). For splenocytes, cells were fixed in 2% paraformaldehyde and permeabilized with 100% methanol on ice, followed by intracellular staining of phosphor-specific antibodies, and monitored by flow cytometry.

### Microscopy

Cell size was measured directly using microscopy. Half million of thymocytes were incubated on the glass slide pre-coated with 0.01% poly-lysine at 37 °C for 10 min. Adherent cell images were captured at indicated time after placing in the hypotonic solution (100 mOsm) using Olympus IX73 inverted microscope. Cell size (area) was quantified from more than 30 cells per time point at each condition using ImageJ software (NIH, https://imagej.nih.gov/ij/).

### Bulk RNA-sequencing

OT-I CD8$^+$ T cells from control and *Lrrc8a* KO mice were stimulated by adding OVA peptides at indicated concentration in freshly isolated splenocytes for 6 h in a humidified, 5% CO$_2$ cell culture incubator. After activation, $1 \times 10^6$ CD8$^+$ Vα2$^+$ OT-I T cells of each treatment were sorted by a FACSArial III (BD Biosciences) directly into TRIzol (Thermo Fisher Scientific) for RNA isolation. Total RNA was extracted using the TRIzol reagent according to the manufacturer's instruction. Each sample group consisted of three biological replicates. The concentration, quality and integrity of RNA samples were determined. cDNA libraries were established and subsequently sequenced on DNBSEQ-T7.

Raw data in FASTQ format were preliminarily filtered by fastp (v0.21.0) software for further analysis. The filtered reads were aligned to the mouse reference genome using HISAT2 (v2.1.0). The read count table was normalized based on the library size factors which generated using DESeq2 (1.34.0) and gene differential expression analysis was performed. Differences in gene expression was considered significant and important if |log2FoldChange| > 1 and padj < 0.05. Principle component analysis was performed on the sample genes and simplifies the complexity of the samples into two-dimensional plot while retaining trends and difference by using plotPCA function in DESeq2 packages. The principal component 1 (PC1) and principal component 2 (PC2) were selected to represent the difference of samples. Linear regression model and generalized additive model (GAM) were performed by changing the parameters (method = "lm" for linear model and method = "gam" for generalized additive model) of geom_smooth function in ggplot2 package. The p-value and correlation were obtained from the output of geom_smooth function. Kyoto Encyclopedia of Genes and Genomes (KEGG) pathways enrichment analysis were performed using ClusterProfiler (v3.18.1) package. Significantly enriched KEGG pathways were identified if padj <0.05. The sequencing data are available at the NCBI Gene Expression Omnibus data base (GEO) under the accession number GSE243985.

### TCR-sequencing

Fresh thymocytes were isolated from *Lrrc8a$^{f/f, Cd4-Cre}$* mice and their littermate control mice (*Lrrc8a$^{f/f}$*), and total RNA was extracted using Takara MiniBEST Universal RNA Extraction Kit (Takara) according to the manufacturer's instruction. TCR library construction and sequencing were previously described. Briefly, TCR library was prepared by three rounds of PCR. The first PCR selectively amplified the total first-strand cDNA by P5-TSO1 primer (10 μM) and mouse TCRAC/BC/DC/GC primer-1 (10 μM) in a 50 μL reaction volume, A 3 min denaturation at 95 °C was followed by 18 cycles of 15 s at 95 °C, 15 s at 65 °C and 25 s at 72 °C, as well as a final extension at 72 °C for 5 min. 1 μL of PCR products from the first round were further amplified by TCR-forward primer (10 μM) and mouse TCRAC/BC/DC/GC primer-2 (10 μM) in a 50 μL reaction volume with the following program: 3 min denaturation at 95 °C, 20 cycles of 15 s at 95 °C, 15 s at 60 °C and 25 s at 72 °C, and final extension at 72 °C for 5 min. For the third round of PCR, 1 μL of PCR products from the second round were subsequently amplified by TCR-forward primer (10 μM) and TCR Reverse primer (10 μM) in a 50 μL reaction volume with the same PCR program as the second round. PCR products were then purified and isolated at approximately 700 bp by VAHTS DNA Clear Beads (Vazyme). Sequencing was performed on the Illumina NovaSeq 6000 platform with the PE150 mode. Sequences were exported from the fluorescent images according to the Illumina data processing pipeline. The NGS data in this work are available in NCBI Sequence Read Archive (GSE243574).

### TCR-seq data processing and analysis

The original data obtained from high-throughput sequencing were converted to raw sequence reads by base calling, and the results were stored in FASTQ format. Low-quality reads and reads without primers were discarded with *trimmomatic* (v0.39). PCR and sequencing errors

were corrected by unique molecular identifiers (UMIs). The reads with same UMI were considered as one TCR clone. TCR information that includes clone count, clone fraction, target sequence, CDR3 amino acid and so on, was obtained by *trust4* (v1.0.6, Song, Nat Meth, 2021). All samples select the same TCR count reading number in normalization with *seqkit* (2.1.0). Further data analysis of the TCR repertoire is performed by the RStudio software, and the results were plotted using the ggplots2 package (v3.3.6) in R language (v4.1.0) (see detailed methodology of TCR repertoire analysis in supplemental information). *p*-value lower than 0.05 (two-sided) was considered as statistically significant.

### Statistical analysis

GraphPad Prism software V8 was used for the plots and statistical analysis (GraphPad Software Inc.). No data were excluded. The statistical details for each experiment are available in the figure legends. All experiments were performed at least two times independently, usually 3 or more, as indicated in the figure legends. n represents the number of animals used, the number of technical replicates, or the number of pictures quantified, as specified in the legends. Data are means ± s.e.m. Statistical analyses were performed using one-way ANOVA, two-way ANOVA or Student's *t* test (two-sided). *P* values of less than 0.05 were considered to be statistically significant.

### Reporting summary

Further information on research design is available in the Nature Portfolio Reporting Summary linked to this article.

## Data availability

Datasets generated by RNA and TCR sequencing in this study have been deposited in the NCBI Gene Expression Omnibus (GEO) database under the accession number GSE243985 and GSE243574. All other data are available in the article and its Supplementary files or from the corresponding author upon request. Source data are provided with this paper.

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

## Acknowledgements
The authors thank Dr. Liangkai Chen for the assistance of statistical analysis. This work was supported by grants from National Natural Science Foundation of China to N. Wu (32370894), the National Key Research and Development Program of China to N. Wu (2021YFC2701200 and 2022YFF0710600).

## Author contributions
Y.W., Z.L., L.Y., and N.W. conceived the project, designed the experiments, and wrote the paper. Y.W. conducted the experiments. Z.S. and B.H. performed the TCR sequencing and analysis. J.P. and J.T. helped for in vitro and in vivo experiments. T.C. and Q.Z. assisted in the RNA sequencing data analysis. S.Y. aided in the LCMV animal experiments. Z.T., X.L., Y.L., R.H., and X.H. helped in the interpretation of results. N.W. supervised the project.

## Competing interests
The authors declare no competing interests.
