## [Peer Review File · Nature Communications]

Cell volume regulated by LRRC8A-formed volume-regulated anion channels controls T cell activation and functionREVIEWER COMMENTS

Reviewer #1 (Remarks to the Author):

Main findings of the manuscript are that inhibition of volume-regulated anion channels (VRAC) or conditional deletion of LRRC8A as one VRAC component impairs TCR-dependent signaling, transcription and T cell activation. This is especially evident at lower peptide concentrations and has direct consequences for anti-viral immunity.

I must congratulate the authors for the well-performed experiments and well-written manuscript. It is easy to follow. The findings advance our understanding on VRAC-mediated control of T cell activation. What I find particularly interesting but remains unanswered is the missing importance at higher peptide concentrations. Thus, in the future it will be important to identify which compensatory mechanisms are operational at such higher peptide concentrations.

Specific comments

1. I am bit puzzled by the differences between effect on cell volume (Extended figure 2i) and forward / sideward scatter (Figure 2c) of LRRC8A ko cells. I would have expected a similar effect. Also, the effect of the LRRC8A ko on T cell signaling and activation is more evident at lower peptide concentrations. However, for FSC the effect is mor evident at higher peptide concentrations. This implies that both effects are unrelated. Also, in the discussion the authors state that “unlike cell size regulation in steady state, the RVD process functions after osmotic swelling, implying LRRC8A-formed VRAC controls T cell signal during T cell swelling after activation, but not before”. If this is the case and LRRC8A controls RVD, it should interfere with direct TCR-mediated signaling such as shown in Figure 7. This aspect needs additional clarification.
2. The effect of LRRC8A ko on CD69 up-regulation (Figure 2b) is rather minor. Activation-induced CD69 up-regulation is a transient process and is subject of temporal differences. Are these differences evident at other timepoints?
3. The authors mention several times the relevance for T cell activation-induced anabolism without measuring metabolic activity or produced metabolites. I would refrain from stating such a relevance.
4. In Figure 5 the relevance of a LRRC8A ko for in vivo viral clearance and formation of

memory T cells is described. The latter was studied at a very early timepoint (14 days). As transfer of LRRC8A ko T cells leads to a delay of viral clearance analysis of memory T cells should be performed at a later timepoint (45 days). It is important to understand whether the effect determined at 14 days (lower MPEC numbers) is caused by a still ongoing acute effector response or due to failure of memory T cell formation.

Minor comments

1. This first sentence on page 11 should be rewritten.

Reviewer #2 (Remarks to the Author):

NCOMMS-23-07017-T “Cell volume regulation during T lymphoblast controls T cell activation and function”

The authors show, that LRRC8A, an obligatory subunit of VRACs is involved in T cell activation and function, especially under weak TCR stimulation. Cell volume regulation mediated by LRRC8A is necessary for proper T cell activation, antiviral immunity and effector-memory transition of (CD8) T cells. Even though the role of VRAC for T cell development and function is quite controversial and puzzling, the authors found a niche to show the impact of the LRRC8A subunit on CD8 and CD4 T cells. Therefore, the experimental design is appropriate for the underlying questions.

However, the data presentation is often too confusing for the reader. Mouse strains (OT-I, OT-II, B6) and cell types (CD4 and CD8) are randomly put together without a suitable explanation, why the authors focus sometimes on both, CD4 and CD8, and sometimes only data for CD8 is shown. By using the different mouse lines in Figure 1a and b, the authors show that they have all tools available; it is unclear why data are shown in such an elusive way.

To enhance the clarity and rigor of the paper the authors should improve to following points:

1. The discrimination between CD4 and CD8 in the experimental setups and the presentation of data should be more stringent. It is not clear, why the authors analyze total T cells for some experiments, for others there is a separation between CD4 and CD8 and

sometimes only CD8 is shown. It is a bit elusive to show some results only with CD8, but on the other hand claim a general T cell effect. The analyzed cell type should be assigned to every Figure. The authors should either focus on only CD8 T cells for their story or show additional results on CD4 T cells, i.e. repeat the experiments also with OT-II.

2. TCR signal strength and affinities: Why is the activation of Nur77 lower in Q4 stimulated cells than in V4 and G4? Obviously, the stimulation with Q4 is not sufficient in the control cells. The authors should check for this discrepancy to validate the data. The authors claim, “more pronounced differences were observed by decreasing the peptide affinity” (p.6, 19). However, why was only N4 used for the analysis of TCR downstream targets in Fig. 3i-l? These experiments should be repeated also with the lower affinity peptides. Vice versa, N4 needs to be included in Fig. 2g.

3. The statistics in this manuscript are mostly too superficial for the experimental design. Especially with such a low number of replicates, the authors have to perform the statistical analysis carefully. Statistical analysis calculated on technical replicates or quantified images, as described in the methods, is not reliable.

a. To strengthen the dataset, the authors have to increase the numbers of independent biological replicates to at least an N=6-9

b. The number of biological replicates needs to be stated for every experiment.

c. Assign the statistical analysis correctly to the respective values within the graph, especially for the dose-response curves e.g. in Fig. 1c, 1e, 1i, Extended Data Fig 1f, Fig. 2b, 2c, ...

d. Post-hoc testing is required, especially for the dose-response curves.

e. A one way ANOVA is not the appropriate statistical test for most of the experiments, since more factors are involved.

4. Analysis of SSC in addition to FSC is not necessary and doesn't have any benefit for the story and data interpretation. Using only FSC is sufficient for the analysis of cell size. This would also be beneficial for the presentation of data and size of the figures.

Minor points:

- 1) Fig. 4f: include the group definitions for I, II and III in the figure legends. If the assigned groups for 4f and 4i are the same, please use the same labelling of groups
- 2) What are Lrr8aFL cells/mice shown in Ext. Fig. 4g? It is not described in the main text, methods or figure legends
- 3) Include the method of calculations for the delta in CD25, CD71 and CD98 shown in Figure 7 and Ext. Fig. 7 in the Figure legends
- 4) Please consider including a glossary for the abbreviations in the manuscript
- 5) Implementing a scheme with the proposed mechanism and alterations upon deletion of LRRC8A

Reviewer #3 (Remarks to the Author):

This paper investigates the role of LRRC8A, a major component of the VRAC involved in the regulation of cell volume by regulated volume decrease. The authors present sound and comprehensive data supporting the notion that LRRC8A plays a role in the early events of TCR signalling and appears to reduce the TCR signal strength. This is observed in vitro and also shown to reduce T cell responses to acute LCMV infection in vivo. Further, very interesting gene regulation patterns of LRRC8A ko T cells in response to antigen stimulation are presented, with some unique patterns of low affinity stimulated ko T cells.

Deletion of LRRC8A results in reduced TCR signalling and consequently T cell activation under standard stimulation conditions. Due to the known function of LRRC8A in cell size regulation and because artificial cell size increase via hypotonic environment enhanced this inhibition, the authors' interpretations focus heavily on this particular feature. However, for reasons more detailed below I'm not fully convinced that the inhibition observed is mainly due to the role of LRRC8A in cell size regulation. Nonetheless the data are strong, and to me point to a much more interesting and potentially novel additional function of LRRC8A that is cell volume regulation independent.

The effect of LRRC8A deletion on early events of T cell activation is demonstrated quite nicely and very clearly.

The effect of either DCPIB or genetic deletion on cell volume seems to be quite marginal though and reduction of cell size also seems to be counterintuitive. The authors show an increase in cell size in a hypotonic milieu, so why does removal of cell size control by inhibition/deletion of LRRC8A result in smaller and not larger cells after antigen stimulation? And how does the reduced cell size (even if only mildly) conform with the notion of cell size control as a mechanism for the reduced T cell responses? Have the authors considered that other mechanisms than RVD are operating here? The fact that differences in TCR signalling can already be detected at the very early events after TCR signalling (minutes) as shown in Fig 7 would suggest a direct inhibitory effect on TCR signal transmission, which to my knowledge would be a novel finding and consistent with VRAC activity being dispensable for T cell development and function (Platt C, JACI 2017, ref 31).

Specific comments / questions:

The authors first confirm that T cells undergo cell growth under their stimulation conditions in vitro and in vivo and then go on to show very nicely that the VRAC inhibitor DCPIB reduces T cell activation.

The authors then introduce a T cell specific knockout of LRRC8A, which is a very nice model and in this reviewer's view a much better system to study the function of LRRC8A than inhibition with DCPIB. While its effect on VRACs is well established (although this hasn't been referenced, can the authors please add?), there have been several reports on off-target effects by DCPIB (ie Afal A, Physiological Reports 2019, and others). What does DCPIB data show that isn't also (and more specifically) demonstrated with the LRRC8A KO T cells? I wonder whether Fig 1 (or the majority of it) could be removed altogether?

Using the LRRC8A ko cells the authors confirm the findings of the DCPIB experiments, very clearly demonstrating a reduction in T cell activation. Expression data for CD69 and CD25 very nicely show a shift in the dose response curve, ie higher levels of the antigen are required to achieve the same outcome. I don't specifically see that this is increased at the lower levels of antigen concentrations as the slope of the curves seems very similar in WT and ko cells and the same plateau is reached eventually.

The restoration of LRRC8A via retroviral transfection provides a great system to separate cell volume regulation from TCR signalling events. Were the retrovirally infected cells restimulated via TCR or mitogens prior to intracellular cytokine staining? Did the authors measure any other functional outcomes that do not require restimulation, ie did the proliferation rates change after LRRC8A restoration?

page 7 line 27: "Intriguingly, the low TCR signal suppressed a plethora of gene expression (group II), mainly in WT OT-I CD8+ T cells but not LRRC8A-deficient cells, implying LRRC8A confers the shutdown of their transcription upon activation (Fig. 4f)."

I don't fully understand what this is saying. Could the authors please clarify? Group II comprises of genes that are downregulated by TCR signalling and this downregulation is dependent on signal strength (the stronger the signal the greater the downregulation), hence reduced downregulation of these genes is consistent with LRRC8A deficiency reducing signal strength in a similar manner as upregulation of group I genes is decreased.

page 7 line 29: I found group III the most intriguing one, as it appeared to be the only one where LRRC8A deletion induced a trend that is different to just reduction of TCR signal strength. Do the authors have more information on this group of genes? Do they correlate with any particular function? Likewise, the group of genes mostly regulated by weak signalling (and further enhanced by LRRC8A deficiency) are most interesting and would warrant some more discussion.

In the LCMV Armstrong infections, given the reduced immune response observed early, has the infection been fully cleared by day 14 in the LRRC8A ko group (virus load on day 14)? Or could ongoing infection and therefore antigen presentation affect the phenotype of the T cells? In Fig 5g there are about 20% of CD44^{low} T cells in the LRRC8A ko group, have these cells undergone activation at all? Have they divided and contributed to the clearance of the infection? Were the P14 CFSE or CTV labelled before transfer to assess whether have undergone cell division?

Assuming the CD44^{low} cells represent non activated T cells, how would the data look like if they were excluded from analysis (gated out)? Would there still be a difference in the phenotype of those cells that have undergone activation?

page 10, line 22: “We observed attenuated CD25 induction upon TCR stimulation in 23 WT OT-I CD8+ T cells under hypotonicity compared with isotonicity (Δ CD25=CD25Iso-CD25Hypo).”

In the experiments conducted with hypotonic conditions culture media was diluted with H₂O, hence that medium did not only have lower osmolarity, but also reduced nutrient content. What effect does that reduced nutrient content have on TCR signalling? Could that be tested by mixing the 300mOsm will saline to retain osmolarity but achieve same nutrient dilution as the 150mOsm solution?

page 11, line 6: “ we observed a similar level of total tyrosine phosphorylation between WT and LRRC8A KO cells in isotonic buffer, indicating LRRC8A is not directly involved in the control of TCR signaling before cell blast”

I don't fully understand this conclusion, particularly in light of data presented in Fig 7g-i, where a small reduction of pLCK, pZAP70 and pLAT is shown (but not commented on) even in isotonic conditions. How does that fit with the smaller cell size of ko vs WT in isotonic conditions? Could the authors please elaborate?

To this reviewer, this would be further evidence of an additional cell volume independent effect of LRRC8A, that affects very early events in the TCR signalling pathways.

Minor comments:

Several figures, but in particular Figure 1 are overly busy.

Some figure legends lack some important details. For example, it is often not clear at which time points the cells were harvested (i.e. Fig 1g,h).

It is curious to me that in the swelling assay in Ext Fig 2i, ko cells reach a plateau and don't continue to grow larger. Is that because larger cells have bursted and are therefore not included in the analysis or does suggest an additional mechanism controlling swelling independent of VRAC?

page 11, line 1: “However, no evident expression of these molecules was unaltered when

WT and LRRC8A KO cells were immersed in a hypotonic buffer”

missing word?

page 11, line 4: is it thymocytes or T cells? At various places throughout the manuscript this not entirely clear.

page 24, line 25: “g-i, Flow cytometry of p-LCKY394 26 (g), p-ZAP70Y319 (h) and p-LATY171 (i). j, p-LCKY394 in OT-I CD8+ T cells stimulated as in g”

I could not find stimulation conditions for ‘g’ in the figure legend. Hence stimulation conditions (and particularly the duration of stimulation) are not clear for panels g-i.

page 13, line 11: “Upon α -CD3 ϵ and α -11 CD28 stimulation, LRRC8C-deficient CD4+ T cells displayed enhanced p53 expression...”

I believe this should read: displayed reduced p53 expression...

We thank the reviewers for their positive comments and helpful advices to further enhance the findings in this study. Below are our responses to the reviewers' comments.

Reviewer #1:

Main findings of the manuscript are that inhibition of volume-regulated anion channels (VRAC) or conditional deletion of LRRC8A as one VRAC component impairs TCR-dependent signaling, transcription and T cell activation. This is especially evident at lower peptide concentrations and has direct consequences for anti-viral immunity. I must congratulate the authors for the well-performed experiments and well-written manuscript. It is easy to follow. The findings advance our understanding on VRAC-mediated control of T cell activation. What I find particularly interesting but remains unanswered is the missing importance at higher peptide concentrations. Thus, in the future it will be important to identify which compensatory mechanisms are operational at such higher peptide concentrations.

Many thanks for your supportive comments!

1) I am bit puzzled by the differences between effect on cell volume (Extended figure 2i) and forward / sideward scatter (Figure 2c) of LRRC8A ko cells. I would have expected a similar effect. Also, the effect of the LRRC8A ko on T cell signaling and activation is more evident at lower peptide concentrations. However, for FSC the effect is more evident at higher peptide concentrations. This implies that both effects are unrelated. Also, in the discussion the authors state that "unlike cell size regulation in steady state, the RVD process functions after osmotic swelling, implying LRRC8A-formed VRAC controls T cell signal during T cell swelling after activation, but not before". If this is the case and LRRC8A controls RVD, it should interfere with direct TCR-mediated signaling such as shown in Figure 7. This aspect needs additional clarification.

Response: This is a critical point in this study. First of all, under the microscope, cells in acute hypotonic solutions exhibit size increase instantly and then decrease due to the RVD, as we observed in Extended Data Fig. 2i. In contrast, the forward scatter, relying on the light passing through the cell detected by flow cytometer, is proportional to the cell size only under isotonic condition (Using flow cytometry, FSC is smaller for LRRC8A KO cells if the experimental settings are the same as in Extended Data Fig. 2i). Besides, the main reason is that the microscope monitored the clear effect of volume increase in response to hypotonicity and the RVD effect of LRRC8A within minutes, whereas the flow cytometry data were obtained after TCR activation under isotonic condition. At this moment, FSC reflects the consequence of VRAC deficiency in T cell activation, but not direct loss of RVD. In addition, the activation markers such as CD25/CD69, as well as the FSC, are both the readouts of T cell activation at different time points, but not the direct volume effect by LRRC8A-mediated RVD, except for Extended Data Fig. 2i. The discrepancy mentioned by the reviewer also may be due to the sensitivity of these two different readouts. CD25/69 are more sensitive to TCR signal strength, but cell size is much less sensitive, requiring strong enough TCR to blast the T cells. Nevertheless, they are coherent.

Beyond that, the puzzling results mentioned by the reviewer are actually two different scenarios but theoretically consistent. For Fig. 2c, the cell size quantified by flow cytometry indicates the T

lymphoblast is compromised in LRRC8A KO cells upon long-term activation, in line with the RVD deficiency due to the loss of LRRC8A. In our model supported by our data, LRRC8A-formed VRAC does not immediately interfere with TCR signaling under isotonic condition, as shown in Fig. 7e, same total tyrosine phosphorylation in WT and LRRC8A KO T cells activated by antibody-crosslinking for only 2 mins. It requires two things: first, the adequate TCR activating signal to augment the intracellular osmolarity; second, long enough time for the augmented osmolarity to drive the cell swelling. During T cell activation, this process is slow, and the cell volume change is relatively small to monitor either by flow cytometer or microscope. However, to uncover the role of cell volume change regulated by LRRC8A in T cell activation, we designed the assays in Figure 7. Loss of LRRC8A reduced T cell's volume control in response to hypotonicity. As a result, enlarged T cells with decreased molecule density exhibit compromised TCR signaling. This mimics that during the slow T cell blast during activation, the compromised cell volume regulation in LRRC8A KO leads to a slightly larger size, resulting in the TCR signaling reduction due to the decrease of molecule density. Consequently, LRRC8A KO T cells fail to be blasted as big as WT cells, as we saw in Fig. 2c. We also have to admit that we could not exclude the cell volume-independent effect of LRRC8A at the current status. It's possible that the LRRC8A-formed channel can be activated during TCR activation/T cell blast to transport molecules required for further T cell activation. To make it clarified, we added a cartoon in Extended Data Fig. 10.

Eventually, we also performed the co-immunoprecipitation assay to verify the point raised by Reviewer 1 that if there is direct interaction between LRRC8A and TCR. Results demonstrated that there is no direct interaction between LRRC8A and TCR β . (Extended Data Fig. 9p). They are discussed on page 12 lines 19-21.

2) The effect of LRRC8A ko on CD69 up-regulation (Figure 2b) is rather minor. Activation-induced CD69 up-regulation is a transient process and is subject of temporal differences. Are these differences evident at other timepoints?

Response: As an early sensitive T cell activation marker, we think that the effect of LRRC8A KO on CD69 is moderate by antibody crosslinking, except for the strong agonist N4 peptide. Moreover, even with N4 at an unsaturated dose, such as 1-10 pM, the difference is about half in LRRC8A KO compared with control cells. In addition, with the decrease of peptide affinity, we observed a much more pronounced effect of LRRC8A on CD69 (in revised Fig. 2h).

As mentioned by the reviewer, CD69 up-regulation is time-dependent. Indeed, we checked the expression of CD69 at different time points. As an immediate early T cell activation marker, the difference for CD69 is most evident at 6 hours post-stimulation. Its expression decreased after that, and the difference diminished as well (Data not shown).

3) The authors mention several times the relevance for T cell activation-induced anabolism without measuring metabolic activity or produced metabolites. I would refrain from stating such a relevance.

Response: Thanks for this kind reminder. We have limited the use of such words in the manuscript.

4) In Figure 5 the relevance of a LRRC8A ko for in vivo viral clearance and formation of memory T cells is described. The latter was studied at a very early timepoint (14 days). As transfer of LRRC8A ko T cells leads to a delay of viral clearance analysis of memory T cells should be performed at a later timepoint (45 days). It is important to understand whether the effect determined at 14 days (lower MPEC numbers) is caused by a still ongoing acute effector response or due to failure of memory T cell formation.

Response: This is a valid point. The LCMV Armstrong infection is an acute infection model. Normally at the viral dose we used, virus were almost cleared at day 7-8 pi already. To answer reviewer's question, we also quantified the virus load at day 14 pi, and no virus were detected in the samples (Results were added to Extended Data Fig. 7c). We also did the experiments for the T cell memory at a later timepoint (day 45 pi). We found that the memory P14 CD8 T cells were significantly fewer in LRRC8A KO than WT controls (revised Fig. 5h). Overall, our data demonstrated that the lower MPEC numbers were due to less efficient activation in LRRC8A KO cells, and the CD8 T cell memory formation was defective in the absence of LRRC8A.

Minor comments

1. This first sentence on page 11 should be rewritten.

Response: We have rewritten this sentence as requested.

Reviewer #2:

NCOMMS-23-07017-T "Cell volume regulation during T lymphoblast controls T cell activation and function"

The authors show, that LRRC8A, an obligatory subunit of VRACs is involved in T cell activation and function, especially under weak TCR stimulation. Cell volume regulation mediated by LRRC8A is necessary for proper T cell activation, antiviral immunity and effector-memory transition of (CD8) T cells. Even though the role of VRAC for T cell development and function is quite controversial and puzzling, the authors found a niche to show the impact of the LRRC8A subunit on CD8 and CD4 T cells. Therefore, the experimental design is appropriate for the underlying questions.

Thanks!

However, the data presentation is often too confusing for the reader. Mouse strains (OT-I, OT-II, B6) and cell types (CD4 and CD8) are randomly put together without a suitable explanation, why the authors focus sometimes on both, CD4 and CD8, and sometimes only data for CD8 is shown. By using the different mouse lines in Figure 1a and b, the authors show that they have all tools available; it is unclear why data are shown in such an elusive way.

Response: Thanks for the reviewer's suggestion. We tried to ameliorate the text in the revised version. Indeed, our data clearly indicated the effect of LRRC8A on T cell activation exists both in CD4⁺ (In revised version, data for CD4⁺ T cells are in Fig. 1f, Fig. 2f,g, Extended Data Fig. 1a-c, f-j, Extended Data Fig. 3a,b,e, Extended Data Fig. 4a-e, Extended Data Fig. 9b,i,l,n) and CD8⁺ T cells. Moreover, this effect applies to both antigen-specific (OT-I, OT-II, P14) and antigen-

independent (anti-CD3, ConA) activation of T cells. We demonstrated the results side by side for CD4 and CD8, antigen-specific or not. Using OT-I and P14 systems, the results were for CD8⁺ T cells, and OT-II mice showed the effect of LRRC8A in CD4⁺ T cells. Thus we believe that the effect of LRRC8A applies to both CD4⁺ and CD8⁺ T cells.

To enhance the clarity and rigor of the paper the authors should improve to following points:

1) The discrimination between CD4 and CD8 in the experimental setups and the presentation of data should be more stringent. It is not clear, why the authors analyze total T cells for some experiments, for others there is a separation between CD4 and CD8 and sometimes only CD8 is shown. It is a bit elusive to show some results only with CD8, but on the other hand claim a general T cell effect. The analyzed cell type should be assigned to every Figure. The authors should either focus on only CD8 T cells for their story or show additional results on CD4 T cells, i.e. repeat the experiments also with OT-II.

Response: Thank you for the advice. We adjusted our statements and clarified the experiments for CD4 and CD8, and OT-I and OT-II as well. We paid attention on the presentation of data and conclusions, and focus on CD8⁺ T cells in the manuscript. OT-II system is in our future experimental plan to address the effect on Th cells.

2) TCR signal strength and affinities: Why is the activation of Nur77 lower in Q4 stimulated cells than in V4 and G4? Obviously, the stimulation with Q4 is not sufficient in the control cells. The authors should check for this discrepancy to validate the data. The authors claim, “more pronounced differences were observed by decreasing the peptide affinity” (p.6, 19). However, why was only N4 used for the analysis of TCR downstream targets in Fig. 3i-l? These experiments should be repeated also with the lower affinity peptides. Vice versa, N4 needs to be included in Fig. 2g.

Response: Due to the different affinities for N4, Q4, V4, and G4, we used different concentrations of peptides in stimulation for Nur77 in Fig. 3a-d. It looks like Nur77 is lower in Q4, because the peptide concentration is 100 pM, compared to 100 nM for V4 and 1000 nM for G4. Regarding the other mutant peptides in TCR signaling and activation, we added lower affinity peptides data in Extended Data Fig. 5.

3) The statistics in this manuscript are mostly too superficial for the experimental design. Especially with such a low number of replicates, the authors have to perform the statistical analysis carefully. Statistical analysis calculated on technical replicates or quantified images, as described in the methods, is not reliable.

- a. To strengthen the dataset, the authors have to increase the numbers of independent biological replicates to at least an N=6-9*
- b. The number of biological replicates needs to be stated for every experiment.*
- c. Assign the statistical analysis correctly to the respective values within the graph, especially for the dose-response curves e.g. in Fig. 1c, 1e, 1i, Extended Data Fig 1f, Fig. 2b, 2c, ...*
- d. Post-hoc testing is required, especially for the dose-response curves.*
- e. A one way ANOVA is not the appropriate statistical test for most of the experiments, since more factors are involved.*

Response: Thanks for your kind reminder. We consulted the statistician and verified all the statistical analyses carefully, and corrections to all the points raised by the reviewer have been made accordingly in the manuscript. We performed all the experiments independently for at least twice, most of them were more than three time. Each time we made triplicates and all the experiments were highly reproducible whatever the experimental settings. Specifically, we summarized all core experimental results with biological replicates $N > 6$ in revised Fig. 2c, and Extended Data Fig. 3a-d.

4) Analysis of SSC in addition to FSC is not necessary and doesn't have any benefit for the story and data interpretation. Using only FSC is sufficient for the analysis of cell size. This would also be beneficial for the presentation of data and size of the figures.

Response: We acknowledge the reviewer for this advice. We have removed the parts containing SSC in the manuscript.

Minor points:

1) Fig. 4f: include the group definitions for I, II and III in the figure legends. If the assigned groups for 4f and 4i are the same, please use the same labelling of groups

Response: Modifications made as requested by the reviewer.

2) What are Lrr8aFL cells/mice shown in Ext. Fig. 4g? It is not described in the main text, methods or figure legends

Response: We added the description for LRRC8A FL in the main text, as well as the figure legends.

3) Include the method of calculations for the delta in CD25, CD71 and CD98 shown in Figure 7 and Ext. Fig. 7 in the Figure legends

Response: Methods were included in the Figure legends.

4) Please consider including a glossary for the abbreviations in the manuscript

Response: All the abbreviations were explained throughout the manuscript.

5) Implementing a scheme with the proposed mechanism and alterations upon deletion of LRRC8A

Response: A cartoon added to illustrate the proposed mechanism in Extended Data Fig. 10.

Reviewer #3:

This paper investigates the role of LRRC8A, a major component of the VRAC involved in the regulation of cell volume by regulated volume decrease. The authors present sound and comprehensive data supporting the notion that LRRC8A plays a role in the early events of TCR signalling and appears to reduce the TCR signal strength. This is observed in vitro and also shown to reduce T cell responses to acute LCMV infection in vivo. Further, very interesting gene

regulation patterns of LRRC8A ko T cells in response to antigen stimulation are presented, with some unique patterns of low affinity stimulated ko T cells.

Deletion of LRRC8A results in reduced TCR signalling and consequently T cell activation under standard stimulation conditions. Due to the known function of LRRC8A in cell size regulation and because artificial cell size increase via hypotonic environment enhanced this inhibition, the authors' interpretations focus heavily on this particular feature. However, for reasons more detailed below I'm not fully convinced that the inhibition observed is mainly due to the role of LRRC8A in cell size regulation. Nonetheless the data are strong, and to me point to a much more interesting and potentially novel additional function of LRRC8A that is cell volume regulation independent.

The effect of LRRC8A deletion on early events of T cell activation is demonstrated quite nicely and very clearly.

Thanks for your supportive comments! In our study, we found clear functional involvement of LRRC8A in T cell activation, at least partially via cell volume regulation, if not fully. We could not exclude the VRAC-independent mechanisms by LRRC8A at this moment.

The effect of either DCPIB or genetic deletion on cell volume seems to be quite marginal though and reduction of cell size also seems to be counterintuitive. The authors show an increase in cell size in a hypotonic milieu, so why does removal of cell size control by inhibition/deletion of LRRC8A result in smaller and not larger cells after antigen stimulation?

Response: Great and critical point! Similar to reviewer 1's first point, cells swelled bigger in the acute hypotonic treatment due to the loss of LRRC8A (Extended Data Fig. 2i). However, in the scenario of T cell activation, T cell blast becoming measurable by flow cytometry is a fairly long-term effect, requiring the intracellular osmolarity increase. The more comprehensive explanation can be found in our response to Reviewer 1's first point.

And how does the reduced cell size (even if only mildly) conform with the notion of cell size control as a mechanism for the reduced T cell responses? Have the authors considered that other mechanisms than RVD are operating here? The fact that differences in TCR signalling can already be detected at the very early events after TCR signalling (minutes) as shown in Fig 7 would suggest a direct inhibitory effect on TCR signal transmission, which to my knowledge would be a novel finding and consistent with VRAC activity being dispensable for T cell development and function(Platt C, JACI 2017, ref 31).

Response: We are delighted that the reviewer pointed out that LRRC8A may function through mechanisms other than VRAC. It's been reported that multiple small molecules go through the LRRC8A-formed channel, including cGAMP, to activate STING signaling. However, we did not observe any activation of the STING pathway when T cells were activated *in vitro* (data not shown). In our results, the amount of TCR signaling molecules was the same between WT and LRRC8A KO (Fig. 7c), and acute TCR signal (crosslinking by anti-CD3 +/- anti-CD28) within 2 minutes was identical as well in WT and LRRC8A KO cells under isotonic conditions (Fig. 7e). This argues that the effect of LRRC8A on T cell activation is subsequent to immediate TCR signal and no direct interaction between LRRC8A and TCR signal transmission. To further validate it, we

performed the co-immunoprecipitation assays, and the results support no direct link between LRRC8A and TCR β (See Extended Data Fig. 9p). Of course, we have not excluded the possibility of mechanisms other than RVD mediated by LRRC8A later after TCR activation. Finally, our data are consistent with the reference mentioned by the reviewer, that there is minimal requirement of LRRC8A for T cell development. However, LRRC8A is definitely involved in forming TCR repertoire in the thymus (Fig. 6).

Specific comments / questions:

1) The authors first confirm that T cells undergo cell growth under their stimulation conditions in vitro and in vivo and then go on to show very nicely that the VRAC inhibitor DCPIB reduces T cell activation.

The authors then introduce a T cell specific knockout of LRRC8A, which is a very nice model and in this reviewer's view a much better system to study the function of LRRC8A than inhibition with DCPIB. While its effect on VRACs is well established (although this hasn't been referenced, can the authors please add?), there have been several reports on off-target effects by DCPIB (ie Afal A, Physiological Reports 2019, and others). What does DCPIB data show that isn't also (and more specifically) demonstrated with the LRRC8A KO T cells? I wonder whether Fig 1 (or the majority of it) could be removed altogether?

Response: We appreciated the reviewer's suggestions on the off-target effects of DCPIB. As the reviewer mentioned, DCPIB is a well-established drug to study the role of VRAC. That is the reason we began with DCPIB. As one of the VRACs, we found similar but smaller phenotypes for LRRC8A KO than DCPIB on T cells. Both pharmacological and genetic evidence solidify the essential role of VRACs in T cell activation.

Further genetic deletion of LRRC8A proves the importance of LRRC8A-formed VRACs in T cells, but does not exclude other VRACs. Thus we believe the DCPIB results would be helpful for our current manuscript and also for other uncharacterized VRACs in the future. Moreover, related references were added into the revised manuscript.

2) Using the LRRC8A ko cells the authors confirm the findings of the DCPIB experiments, very clearly demonstrating a reduction in T cell activation. Expression data for CD69 and CD25 very nicely show a shift in the dose response curve, ie higher levels of the antigen are required to achieve the same outcome. I don't specifically see that this is increased at the lower levels of antigen concentrations as the slope of the curves seems very similar in WT and ko cells and the same plateau is reached eventually.

The restoration of LRRC8A via retroviral transfection provides a great system to separate cell volume regulation from TCR signalling events. Were the retrovirally infected cells restimulated via TCR or mitogens prior to intracellular cytokine staining? Did the authors measure any other functional outcomes that do not require restimulation, ie did the proliferation rates change after LRRC8A restoration?

Response: With the antigen affinity decrease, we observed that the difference between WT and LRRC8A KO cells increased at a specific concentration. In other words, LRRC8A-deficient cells required higher concentrations of lower-affinity antigen to get activated.

Retrovirally infected cells were restimulated via TCR by plate-bound anti-CD3 and followed by intracellular cytokine staining. We did not measure other functional outcomes without restimulation. Indeed, retroviral infection was done 24 hours post-activation of purified naïve CD8 T cells by plate-bound anti-CD3 and anti-CD28 (strong enough TCR signal for LRRC8A KO cells). One day after infection, IL-2 was supplemented for the expansion of T cells. In this case, we did not observe any proliferation defect between the cells with or without LRRC8A.

3) page 7 line 27: *“Intriguingly, the low TCR signal suppressed a plethora of gene expression (group II), mainly in WT OT-I CD8+ T cells but not LRRC8A-deficient cells, implying LRRC8A confers the shutdown of their transcription upon activation (Fig. 4f).” I don’t fully understand what this is saying. Could the authors please clarify? Group II comprises of genes that are downregulated by TCR signalling and this downregulation is dependent on signal strength (the stronger the signal the greater the downregulation), hence reduced downregulation of these genes is consistent with LRRC8A deficiency reducing signal strength in a similar manner as upregulation of group I genes is decreased.*

Response: yes, exactly as the reviewer's comprehension, Group II's regulation seems to be the opposite of Group I. We re-phrased the text that the reviewer mentioned above to clarify it. See page 7 line 30-34.

4) page 7 line 29: *I found group III the most intriguing one, as it appeared to be the only one where LRRC8A deletion induced a trend that is different to just reduction of TCR signal strength. Do the authors have more information on this group of genes? Do they correlate with any particular function? Likewise, the group of genes mostly regulated by weak signalling (and further enhanced by LRRC8A deficiency) are most interesting and would warrant some more discussion.*

Response: We agree with the reviewer that Group III is the most interesting. We performed the GO enrichment analysis for the genes in Group III. The GTPase activity, guanyl nucleotide binding, and guanyl ribonucleotide binding pathways are top enriched (See Extended Data Fig. 6b). We adjusted the related text and added some more discussion as suggested by the reviewer (page 7, line 34-36).

5) *In the LCMV Armstrong infections, given the reduced immune response observed early, has the infection been fully cleared by day 14 in the LRRC8A ko group (virus load on day 14)? Or could ongoing infection and therefore antigen presentation affect the phenotype of the T cells? In Fig 5g there are about 20% of CD44low T cells in the LRRC8A ko group, have these cells undergone activation at all? Have they divided and contributed to the clearance of the infection? Were the P14 CFSE or CTV labelled before transfer to assess whether have undergone cell division? Assuming the CD44low cells represent non activated T cells, how would the data look like if they were excluded from analysis (gated out)? Would there still be a different in the phenotype of those cells that have undergone activation?*

Response: similar to reviewer 1's concern, in our experimental settings, virus was almost eliminated by the day 7-8 pi. In addition, we checked the virus load at day 14, as shown by undetected virus in revised Extended Data Fig. 7c. Regarding the CD44low T cells mentioned by the reviewer, we labeled the P14 cells with CTV before the transfer, and found that all the P14

cells recovered at day 3 post-infection have undergone cell division, and CD44^{hi} cells were identical in WT and LRRC8A KO (Extended Data Fig. 7a,b). In addition, CD44^{low} P14 cells recovered at day 14 pi were all CTV negative (Extended Data Fig. 7d). Thus CD44^{low} population has been activated during infection, but possibly lost its expression in the absence of LRRC8A later. We agree with the reviewer that the CD44^{low} T cells in LRRC8A KO at day 14 are interesting phenotype and worthy of pursuing in our future study.

6) page 10, line 22: *“We observed attenuated CD25 induction upon TCR stimulation in 23 WT OT-I CD8+ T cells under hypotonicity compared with isotonicity ($\Delta CD25 = CD25_{Iso} - CD25_{Hypo}$).” In the experiments conducted with hypotonic conditions culture media was diluted with H₂O, hence that medium did not only have lower osmolarity, but also reduced nutrient content. What effect does that reduced nutrient content have on TCR signalling? Could that be tested by mixing the 300mOsm will saline to retain osmolarity but achieve same nutrient dilution as the 150mOsm solution?*

Response: We performed the following experiments to address the speculation that the diluted nutrients have any effects on TCR signaling in our settings. First, we used PBS to dilute the nutrients to the same concentration as we used in the hypotonic buffer, but without altering the osmolarity. We did not observe the difference of p-LCK for WT and LRRC8A KO cells (Extended Data Fig. 9l,m). Then, we used isotonic and hypotonic solutions by adjusting only mannitol in the buffer (reported in Chun Zhou, Immunity, 2020). In this setting, we excluded entirely the role of nutrients, and we also observed the same results as we saw in cell medium diluted with water (Extended Data Fig. 9n,o). Thus, we can conclude that the signaling reduction in LRRC8A KO cells is due to the decreased osmolarity, not the diluted nutrients. Results were added in the revised Extended Data Fig. 9.

7) page 11, line 6: *“ we observed a similar level of total tyrosine phosphorylation between WT and LRRC8A KO cells in isotonic buffer, indicating LRRC8A is not directly involved in the control of TCR signaling before cell blast” I don't fully understand this conclusion, particularly in light of data presented in Fig 7g-i, where a small reduction of pLCK, pZAP70 and pLAT is shown (but not commented on) even in isotonic conditions. How does that fit with the smaller cell size of ko vs WT in isotonic conditions? Could the authors please elaborate?*

Response: this is a very critical point in our study. Total tyrosine phosphorylation results came from anti-CD3 and anti-CD28 crosslinking for only 2 minutes in Fig. 7e. We believe this immediate TCR activation has not triggered the cell volume increase yet. However, as the reviewer indicated, we also noticed slight decrease of phosphorylation of LCK, ZAP70 and LAT at isotonic condition in Fig 7g-i. We have to emphasize that for the phosphor-flow, we stimulated the cells for at least 30 minutes. It's possible that at this time point, intracellular osmolarity has increased to trigger the cell volume change, and uncovered the effect of LRRC8A. Of course, we cannot fully exclude an additional cell volume independent effect of LRRC8A, but we do observe much greater effect of LRRC8A on T cell signaling under hypotonic condition (Fig. 7e-i). We added a cartoon to summarize our model in Extended Data Fig. 10 and some comments in the manuscript as requested by the reviewer.

Minor comments:

Several figures, but in particular Figure 1 are overly busy. Some figure legends lack some important details. For example, it is often not clear at which time points the cells were harvested (i.e. Fig 1g,h).

Response: Thanks for the advice. We have added more details throughout all the figure legends.

It is curious to me that in the swelling assay in Ext Fig 2i, ko cells reach a plateau and don't continue to grow larger. Is that because larger cells have bursted and are therefore not included in the analysis or does suggest an additional mechanism controlling swelling independent of VRAC?

Response: That is a great point. We chose the osmolarity to avoid cell lysis in the hypotonic buffer, which would not make the cell die within the treatment period (Data not shown). This result may imply that LRRC8A-formed VRAC is the major one in T cells.

page 11, line 1: "However, no evident expression of these molecules was unaltered when WT and LRRC8A KO cells were immersed in a hypotonic buffer" missing word?

Response: the sentence was corrected in the manuscript.

page 11, line 4: is it thymocytes or T cells? At various places throughout the manuscript this not entirely clear.

Response: it is thymocytes on page 11, line 4. We specified the cell types throughout the manuscript.

page 24, line 25: "g-i, Flow cytometry of p-LCKY394 26 (g), p-ZAP70Y319 (h) and p-LATY171 (i). j, p-LCKY394 in OT-I CD8+ T cells stimulated as in g" I could not find stimulation conditions for 'g' in the figure legend. Hence stimulation conditions (and particularly the duration of stimulation) are not clear for panels g-i.

Response: we stimulated the OT-I CD8⁺ T cells by N4 peptide at 1 nM for 30 min, in Fig. 7g-i. All the detailed information has been added in the figure legends.

page 13, line 11: "Upon α -CD3 ϵ and α -CD28 stimulation, LRRC8C-deficient CD4+ T cells displayed enhanced p53 expression..." I believe this should read: displayed reduced p53 expression...

Response: we verified the source reference and corrected this mistake. Actually, total and phosphorylated p53 is decreased in LRRC8C KO cells.

REVIEWERS' COMMENTS

Reviewer #1 (Remarks to the Author):

The authors addressed all my comments adequately.

Reviewer #2 (Remarks to the Author):

NCOMMS-23-07017-T "Cell volume regulation during T lymphoblast controls T cell activation and function"

The authors addressed most major and minor points. They emphasized the focus on CD8+ T cells in their current study. The clarity of the manuscript improved a lot by description of the exact cell type, mouse strain and experimental details in the figure legends or respective results section.

However, I would like to add one minor concern:

Since the size of the figures remains that large, it would still be beneficial to include such details also consistently in all figures (e.g. Fig. 1c/d does not contain an information about the mouse strain; Fig 1e/g/h and k are labelled with OT-I at the histograms). Please consider a consistent annotation within the figures.

Reviewer #3 (Remarks to the Author):

The authors have addressed most of the reviewers comments and accordingly the revised manuscript has increased in clarity and describes an elegant study.

I quite like the addition of Extended Data Fig 10 and would suggest to elevate it to the main Figures.

Remaining concerns / comments:

Page 7 line 30: "On the contrary, the low TCR signal suppressed a plethora of gene expression (group II), mainly in WT OT-I CD8+ T cells but not LRRC8A-deficient cells, implying LRRC8A confers the shutdown of their transcription upon activation (Fig. 4f)."

I'm still unconvinced by this interpretation. The way it is written implies a direct regulation

of LRRC8A on the gene regulation and strongly suggest rephrasing.

This set of genes in group is clearly downregulated by TCR signaling, with the strongest effect seen with the highest affinity and concentration (N4high). The weaker the signal the weaker the downregulation. Hence a reduction of downregulation in the absence of LRRC8A is consistent with weaker stimulation signal (as was shown in Fig 3), but does not necessarily imply that LRRC8A directly confers shutdown of the transcription of these genes but just a consequence of weaker signal being transmitted.

page 11, line 17: I'm still confused about the use of thymocytes here. All other data shown in the figure were done on T cells and there is no explanation given why some experiments were done on thymocytes, nor is particularly mentioned in the text that these are different cells.

Please clarify and specify clearly in the text which cell type was used and why.

Minor comments

Fig legends in Fig 1 incorrect from 1g

Fig 2c, should show MFI instead of %, similar to Fig 1d (% already shown in Fig 2B)

Fig 3e-h: how was the cut off between neg and pos set? It is at around 8000 in panel e and f, and a long way above the background, whereas in panel g and h it is at 2000, just above background. The background seems to be similar in all panels. The gating should be consistent between conditions.

We thank the reviewers for their positive comments to our revised manuscript. Below are our responses to the reviewers' new comments.

Reviewer #1 (Remarks to the Author):

The authors addressed all my comments adequately.

Thank you!

Reviewer #2 (Remarks to the Author):

NCOMMS-23-07017-T "Cell volume regulation during T lymphoblast controls T cell activation and function".

The authors addressed most major and minor points. They emphasized the focus on CD8+ T cells in their current study. The clarity of the manuscript improved a lot by description of the exact cell type, mouse strain and experimental details in the figure legends or respective results section.

Thanks a lot.

However, I would like to add one minor concern: Since the size of the figures remains that large, it would still be beneficial to include such details also consistently in all figures (e.g. Fig. 1c/d does not contain an information about the mouse strain; Fig 1e/g/h and k are labelled with OT-I at the histograms). Please consider a consistent annotation within the figures.

Response: Thanks for your helpful suggestion. We specified the details within the figures (Fig. 1c/d/e/g/h and k, performed with OT-I mice).

Reviewer #3 (Remarks to the Author):

The authors have addressed most of the reviewers comments and accordingly the revised manuscript has increased in clarity and describes an elegant study. I quite like the addition of Extended Data Fig 10 and would suggest to elevate it to the main Figures.

Response: We appreciate a lot for your supportive comments. We also think it is a great suggestion to move the Extended Data Fig 10 to the main figures. Now, it is Figure 8 in the manuscript.

Remaining concerns / comments:

Page 7 line 30: "On the contrary, the low TCR signal suppressed a plethora of gene expression (group II), mainly in WT OT-I CD8+ T cells but not LRRC8A-deficient cells, implying LRRC8A confers the shutdown of their transcription upon activation (Fig. 4f)." I'm still unconvinced by this interpretation. The way it is written implies a direct regulation of LRRC8A on the gene regulation and strongly suggest rephrasing. This set of genes in group is clearly downregulated by TCR

signaling, with the strongest effect seen with the highest affinity and concentration (N4^{high}). The weaker the signal the weaker the downregulation. Hence a reduction of downregulation in the absence of LRRC8A is consistent with weaker stimulation signal (as was shown in Fig 3), but does not necessarily imply that LRRC8A directly confers shutdown of the transcription of these genes but just a consequence of weaker signal being transmitted.

Response: We agree with reviewer 3 that LRRC8A did not directly shut down the genes transcription. It is indeed due to the regulation of LRRC8A on TCR signaling. We have adjusted the sentence on page 8 line 9-10, to avoid this mis-interpretation.

page 11, line 17: I'm still confused about the use of thymocytes here. All other data shown in the figure were done on T cells and there is no explanation given why some experiments were done on thymocytes, nor is particularly mentioned in the text that these are different cells. Please clarify and specify clearly in the text which cell type was used and why.

Response: Thymocytes are immune cells in the thymus before they develop into mature T cells. Almost all thymocytes express surface TCR except the double-negative subsets. This makes them ideal for the study of tyrosine phosphorylation by western blot. Another advantage of thymocytes is the cell amount. One mouse could provide us with sufficient cells for the experiment without extensive manipulations of purification. We clarified it on page 11 line 31-33.

Minor comments

Fig legends in Fig 1 incorrect from 1g

Response: We have checked the figure legends in Fig 1 and made sure they are correct in the newly revised version.

Fig 2c, should show MFI instead of %, similar to Fig 1d (% already shown in Fig 2B)

Response: Fig. 2c is pooled statistical results from different mice, as requested by reviewer 2 in the first revision. Actually, it is obvious that CD69 and CD25 have a digital expression pattern upon TCR signaling. Percentage is more appropriate for their analysis, especially under unsaturated signal strength. The MFI in Fig. 1d intends to show that though the percentage of CD69 and CD25 positive cells are comparable in the vehicle and DCPIB groups, the surface expression intensity differed.

Fig 3e-h: how was the cut off between neg and pos set? It is at around 8000 in panel e and f, and a long way above the background, whereas in panel g and h it is at 2000, just above background. The background seems to be similar in all panels. The gating should be consistent between conditions.

Response: Thanks a lot for the advice. We have adjusted the figures for panels e and f to keep them consistent between conditions and replaced panel e with the data at lower N4 concentration.